# ADBM: Adversarial Diffusion Bridge Model for Reliable Adversarial Purification

**Xiao Li**[1*], **Wenxuan Sun**[1,2*], **Huanran Chen**[1,3], **Qiongxiu Li**[4],
**Yining Liu**[1,5], **Yingzhe He**[6], **Jie Shi**[6], **Xiaolin Hu**[1†]
[1]Department of Computer Science and Technology, Tsinghua University
[2]Peking University  [3]Beijing Institute of Technology  [4]Aalborg University
[5]Harbin Institute of Technology, Weihai  [6]Huawei Technologies
lixiao20@mails.tsinghua.edu.cn, sunwenxuan@stu.pku.edu.cn
huanranchen@bit.edu.cn, qili@es.aau.dk
22S030184@stu.hit.edu.cn, {heyingzhe, shi.jie1}@huawei.com
xlhu@mail.tsinghua.edu.cn

## Abstract

Recently Diffusion-based Purification (DiffPure) has been recognized as an effective defense method against adversarial examples. However, we find DiffPure which directly employs the original pre-trained diffusion models for adversarial purification, to be suboptimal. This is due to an inherent trade-off between noise purification performance and data recovery quality. Additionally, the reliability of existing evaluations for DiffPure is questionable, as they rely on weak adaptive attacks. In this work, we propose a novel Adversarial Diffusion Bridge Model, termed ADBM. ADBM directly constructs a reverse bridge from the diffused adversarial data back to its original clean examples, enhancing the purification capabilities of the original diffusion models. Through theoretical analysis and experimental validation across various scenarios, ADBM has proven to be a superior and robust defense mechanism, offering significant promise for practical applications. Code is available at `https://github.com/LixiaoTHU/ADBM`.

## 1 Introduction

An intriguing problem in machine learning models, particularly Deep Neural Networks (DNNs), is the existence of adversarial examples (Szegedy et al., 2014; Goodfellow et al., 2015). These examples introduce imperceptible adversarial perturbations leading to significant errors, which has posed severe threats to practical applications (Eykholt et al., 2018; Li et al., 2025). Numerous methods have been proposed to defend against adversarial examples. But attackers can still evade most early methods by employing adaptive attacks (Athalye et al., 2018; Tramèr et al., 2020). Adversarial Training (AT) methods (Madry et al., 2018; Li et al., 2024c;b) are recognized as effective defense methods against adaptive attacks. However, AT typically involves re-training the entire DNNs using adversarial examples, which is impractical for real-world applications. Moreover, the effectiveness of AT is often limited to the specific attacks it has been trained against, making it brittle against unseen threats (Zhang et al., 2019; Laidlaw et al., 2021).

Recently, Adversarial Purification (AP) methods (Shi et al., 2021; Yoon et al., 2021) have gained increasing attention as they offer a potential solution to defend against unseen threats in a plug-and-play manner without retraining the classifiers. These methods utilize the so-called purification module, which exploits techniques such as generative models, as a pre-processing step to restore clean examples from adversarial examples, as illustrated in Figure 1(a). Recently, diffusion models (Ho et al., 2020), one type of generative model renowned for their efficacy, have emerged as potential AP solutions (Nie et al., 2022). Diffusion models learn transformations between complex data distributions and simple distributions like the Gaussian distribution through forward diffusion and reverse prediction processes. In the context of adversarial defense, Diffusion-based Purification

---

*Equal contribution.
†Correspondence to: Xiaolin Hu.

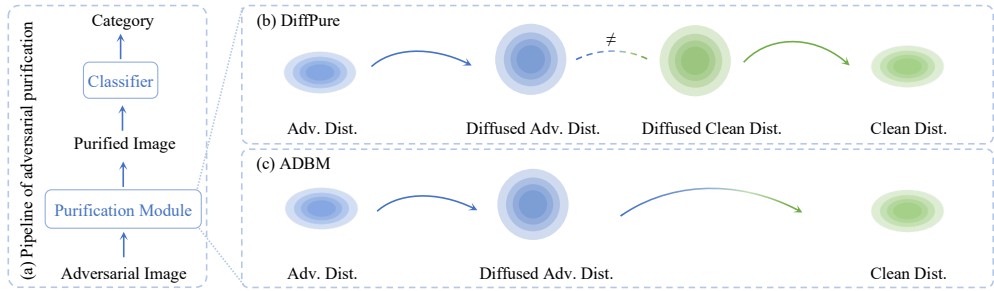

Figure 1: The inference pipeline of AP (a) and the comparison between DiffPure (b) and ADBM (c). DiffPure relies on the diffused adversarial data distribution (Diffused Adv. Dist.) being close enough to the diffused clean data distribution. ADBM directly builds a reverse process from the diffused adversarial data distribution to clean data distribution.

(DiffPure) (Nie et al., 2022), which tries to purify the adversarial examples by first adding Gaussian noise through the forward process with a small diffusion timestep and then recovering clean examples (removing the added Gaussian noise together with adversarial noise) through the reverse process, has achieved superior performance among recent AP methods.

However, we find that utilizing DiffPure in the context of AP is suboptimal. The primary reason is that DiffPure directly employs pre-trained diffusion models originally designed for generative tasks, rather than specifically for AP tasks. To delve deeper, the divergence in data distributions between clean and adversarial examples (Stutz et al., 2019; Xie et al., 2020) often leads to noticeable differences in their respective diffused distributions, while DiffPure relies on the assumption that the two diffused distributions are sufficiently close, such that the original reverse process can recover the diffused adversarial data distribution, as depicted in Figure 1(b). The conflict between the assumption and the actual situation compromises the effectiveness of AP. DiffPure attempts to mitigate this conflict by introducing significant noise with a large diffusion timestep. Nevertheless, this solution is impractical; a substantial diffusion timestep can severely corrupt the global structure of the input, resulting in a fundamental trade-off between noise purification efficiency and recovery quality. In addition, we observe that existing evaluations for DiffPure rely on weak adaptive attacks, which may inadvertently give a false sense of security regarding diffusion-based purification.

To address the aforementioned problems, this study performs a systematic investigation. We first establish a reliable adaptive attack evaluation method for diffusion-based purification. With the rigorous evaluation, our preliminary findings suggest that the robustness of DiffPure is overestimated in existing works (Nie et al., 2022; Lee & Kim, 2023; Chen et al., 2023a). To improve the robustness of diffusion-based purification, we then propose a novel Adversarial Diffusion Bridge Model, termed ADBM. Unlike original diffusion models relying on the similarity between the diffused distributions of clean and adversarial examples for a balanced trade-off, ADBM constructs a direct reverse process (or "bridge") from the diffused adversarial data distribution to the distribution of clean examples, as shown in Figure 1(c). The theoretical analysis supports the superiority of ADBM over using original pre-trained diffusion models (DiffPure). In addition, we discuss how to accelerate ADBM for efficient adversarial purification to enhance the practicality of ADBM as a defense mechanism.

Experimental results show that ADBM achieved better robustness than DiffPure under reliable adaptive attacks. In particular, ADBM achieved a 4.4% robustness gain compared with DiffPure on average on CIFAR-10 (Krizhevsky et al., 2009), while the clean accuracies kept comparable. ADBM also demonstrated competitive performances with AT methods for seen adversarial threats and stood out among recent AP methods. Furthermore, ADBM exhibited much better adversarial robustness than AT methods when facing unseen threats. Additionally, the evaluation results against transfer-based and query-based attacks indicate the practicality of ADBM compared with existing methods.

Our main contributions can be summarized as follows:

- We propose ADBM, a novel adversarial diffusion bridge model, as an efficient AP method. Additionally, we investigate methods to accelerate the proposed ADBM.
- We conduct a theoretical analysis to illustrate the superiority of ADBM.

- We develop a simple yet reliable adaptive attack evaluation method for diffusion-based purification methods.
- Experiments in various settings validate the effectiveness of ADBM, highlighting its reliability and potential as a defense method for practical scenarios when compared with AT.

## 2 PRELIMINARY AND RELATED WORK

**Diffusion models.** Given a data distribution $q(\mathbf{x}_0)$, DDPM (Ho et al., 2020) constructs a discrete-time Markov chain $\{\mathbf{x}_0, \ldots, \mathbf{x}_T\}$ as the forward process for $\mathbf{x}_0 \sim q(\mathbf{x}_0)$. Gaussian noise is gradually added to $\mathbf{x}_0$ during the forward process following a scaling schedule $\{\beta_0, \beta_1, \cdots, \beta_T\}$, where $\beta_0 = 0$ and $\beta_T \to 1$, such that $\mathbf{x}_T$ is near an isotropic Gaussian distribution:

$$q(\mathbf{x}_t|\mathbf{x}_{t-1}) := \mathcal{N}(\mathbf{x}_t; \sqrt{1-\beta_t}\mathbf{x}_{t-1}, \beta_t \mathbf{I}). \tag{1}$$

Denote $\alpha_t := 1 - \beta_t$ and $\bar{\alpha}_t := \prod_{i=1}^{t} \alpha_i$, then

$$q(\mathbf{x}_t|\mathbf{x}_0) = \mathcal{N}(\mathbf{x}_t; \sqrt{\bar{\alpha}_t}\mathbf{x}_0, (1-\bar{\alpha}_t)\mathbf{I}), i.e., \mathbf{x}_t(\mathbf{x}_0, \boldsymbol{\epsilon}) = \sqrt{\bar{\alpha}_t}\mathbf{x}_0 + \sqrt{1-\bar{\alpha}_t}\boldsymbol{\epsilon}, \boldsymbol{\epsilon} \sim \mathcal{N}(\mathbf{0}, \mathbf{I}). \tag{2}$$

To generate examples, the reverse distribution $q(\mathbf{x}_{t-1}|\mathbf{x}_t)$ should be learned by a model. But it is hard to achieve it directly. In practice, DDPM considers the conditional reverse distribution $q(\mathbf{x}_{t-1}|\mathbf{x}_t, \mathbf{x}_0)$ and uses $\mathbf{x}_\theta(\mathbf{x}_t, t)$ as an estimate of $\mathbf{x}_0$, where

$$\mathbf{x}_\theta(\mathbf{x}_t, t) := (\mathbf{x}_t - \sqrt{1-\bar{\alpha}_t}\boldsymbol{\epsilon}_\theta(\mathbf{x}_t, t))/\sqrt{\bar{\alpha}_t}. \tag{3}$$

For a given $\mathbf{x}_0$, the training loss $L_d$ of diffusion models is thus defined as

$$L_d = \mathbb{E}_{\boldsymbol{\epsilon}, t}\left[\|\boldsymbol{\epsilon} - \boldsymbol{\epsilon}_\theta(\sqrt{\bar{\alpha}_t}\mathbf{x}_0 + \sqrt{1-\bar{\alpha}_t}\boldsymbol{\epsilon}, t)\|^2\right]. \tag{4}$$

To accelerate the reverse process of DDPM, which typically involves hundreds of steps, Song et al. (2020) propose a DDIM sampler based on the intuition that the multiple reverse steps can be performed at a single step via a non-Markov process. Song et al. (2021) generalize the discrete-time diffusion model to continuous-time from the Stochastic Differential Equation (SDE) perspective.

Another series of works seemly related to ours are diffusion bridges (Zhou et al., 2024; Zheng et al., 2024; Bortoli et al., 2021; Liu et al., 2023b), which enlarge the design space of diffusion models by interpolating between two paired distributions. However, ADBM and current diffusion bridge models have distinct motivations and designs. Current diffusion bridge models either map the data distribution to a simple prior distribution (Bortoli et al., 2021) or directly create a Gaussian bridge (Zhou et al., 2024) or an optimal transport bridge (Zheng et al., 2024; Liu et al., 2023b) between two data distributions. The former approach (Bortoli et al., 2021) fundamentally cannot be used for AP, while the latter methods (Zhou et al., 2024; Bortoli et al., 2021; Liu et al., 2023b) tend to a theoretical training collapse for the AP task. We give a detailed discussion to these works in Appendix A.1.

**Diffusion models for adversarial robustness.** Recent studies have demonstrated the efficacy of diffusion models (Ho et al., 2020) in enhancing adversarial robustness in several ways. Some researches leverage much data generated by diffusion models to improve the AT performance (Gowal et al., 2021; Wang et al., 2023), but these AT-based methods do not generalize well under unseen threat models. DiffPure (Nie et al., 2022) employs a diffusion model as a plug-and-play pre-processing module to purify adversarial noise. Wang et al. (2022) improved DiffPure by employing inputs to guide the reverse process of the diffusion model to ensure the purified examples are close to input examples. Zhang et al. (2023) improved DiffPure by incorporating the reverse SDE with multiple Langevin dynamic runs. Zhang et al. (2024) maximized the evidence lower bound of the likelihood estimated by diffusion models to increase the likelihood of corrupted images. DiffPure has also shown potential in improving certified robustness within the framework of randomized smoothing (Carlini et al., 2023; Xiao et al., 2023). Nevertheless, the practicality of randomized smoothing is greatly hindered by the time-consuming Monte Carlo sampling (Cohen et al., 2019). Different from these works, we present an diffusion-based purification method for empirical robustness in practical scenarios. In addition to these works, Chen et al. (2023a) show that a single diffusion model can be transformed into an adversarially robust diffusion classifier (RDC) using Bayes' rule. However, compared with diffusion-based purification, RDC requires thousands of times the inference cost and may not scale up, limiting its practical usefulness. We discuss it further in Appendix A.2.

Table 1: Accuracies (%) of DiffPure under various evaluation attacks with an $l_\infty$ bound of $\epsilon_\infty = 8/255$ on CIFAR-10. The timestep $T$ of DiffPure was 100, following the original implementation.

| Evaluation | Clean Acc | Robust Acc |
|---|---|---|
| BPDA (Athalye et al., 2018) | $90.49 \pm 0.97$ | $81.40 \pm 0.16$ |
| Nie et al. (2022) | $90.07 \pm 0.97$ | $71.29 \pm 0.55$ |
| Chen et al. (2023a) | 90.97 | 53.52 |
| Lee & Kim (2023) | $90.43 \pm 0.60$ | $51.13 \pm 0.87$ |
| Ours (EOT=20, steps=200) | | $45.83 \pm 1.27$ |
| Ours (EOT=40, steps=200) | $90.49 \pm 0.97$ | $\mathbf{45.64} \pm 1.14$ |
| Ours (EOT=20, steps=400) | | $46.16 \pm 1.33$ |

Table 2: Accuracies (%) of DiffPure on CIFAR-10 with forward step 100, various reverse steps, and different samplers. The robust accuracy was evaluated under an $l_\infty$ bound of $\epsilon_\infty = 8/255$.

| Reverse | DDPM | | DDIM | |
|---|---|---|---|---|
| Step | Clean | Robust | Clean | Robust |
| 100 | 90.49 | **45.83** | 93.50 | 41.21 |
| 10 | 84.77 | 41.02 | 92.18 | 41.02 |
| 5 | 68.75 | 31.25 | 92.38 | **42.16** |
| 2 | 29.10 | - | 91.79 | 41.02 |
| 1 | 17.58 | - | 91.80 | 41.41 |

## 3 RELIABLE EVALUATION FOR DIFFPURE

Before delving into the details of ADBM, it is important to discuss the white-box adaptive attack for diffusion-based purification first. As the original implementation of DiffPure needs dozens of prediction steps in the reverse process, it is challenging to compute the full gradient of the whole purification process due to memory constraints. To evaluate the robustness of diffusion-based purification, several techniques have been proposed. But we found that the adaptive evaluations for DiffPure remained insufficient, as detailed in Appendix B.1. We build on these previous insights (Nie et al., 2022; Chen et al., 2023a; Lee & Kim, 2023) to develop a straightforward yet effective adaptive attack method against diffusion-based purification. We employ the gradient-based PGD attack (Madry et al., 2018), utilizing the full gradient calculation via gradient-checkpointing and incorporating a substantial number of EOT (Athalye et al., 2018) and iteration steps. Especially, we set the PGD iteration steps to 200, with 20 EOT samples for each iteration. We note that the cost of reliable evaluation is high yet worthwhile to avoid a false sense of security on DiffPure, as discussed in Appendix B.2. Tab. 1 shows the results of DiffPure with a DDPM sampler under different evaluations.[1] Consistent with Nie et al. (2022), we conducted the adaptive attack three times on a subset of 512 randomly sampled images from the test set of CIFAR-10. The results demonstrate that our attack significantly reduced the reported robustness of DiffPure, lowering it from 71.29% and 51.13% to 45.83% when compared with the originally reported results (Nie et al., 2022) and the recent best practice (Lee & Kim, 2023), respectively. We found that increasing the number of iterations or EOT steps further did not lead to higher attack success rates.

With the reliable evaluation method, we further investigated the influence of reverse steps and forward steps on the robustness of DiffPure. This exploration offers insights into the effectiveness of diffusion-based purification methods.

**Reverse steps.** DiffPure suffers from a high inference cost, which greatly hinders its practical application. The reverse step directly impacts the inference cost of DiffPure. Following Lee & Kim (2023), we evaluated both the original DDPM sampler and the DDIM sampler. We present the results in Tab. 2 using different reverse steps and samplers (standard deviation omitted, about 1%). The results show that decreasing the number of reverse steps significantly reduced both clean accuracy and robustness when using DDPM. However, for the DDIM sampler, although clean accuracy slightly decreased with fewer reverse steps, robustness was not significantly affected. This can be attributed to the fact that DDIM does not introduce additional randomness during the reverse process.

**Forward steps.** We observed a continuous decrease in clean accuracy and a continuous increase in robustness as the number of forward steps increased. Details can be found in Appendix B.3.

## 4 ADVERSARIAL DIFFUSION BRIDGE MODEL

ADBM aims to construct a reverse bridge directly from the diffused adversarial data distribution to the clean data distribution. We derive the training objective for ADBM in Sec. 4.1, and explain

---

[1]The DDPM sampler represents one discretization version of the reverse VP-SDE sampler (Song et al., 2021), which is originally employed in DiffPure. Their differences become negligible at a large timestep, allowing us to use VP-SDE and DDPM interchangeably throughout our work.

how to obtain the adversarial noise for training ADBM in Sec. 4.2. The AP inference process using ADBM is described in Sec. 4.3. We finally show that ADBM has good theoretical guarantees for AP.

## 4.1 TRAINING OBJECTIVE

ADBM is a diffusion model specifically designed for purifying adversarial noise. It adopts a forward process similar to DDPM, with the difference that ADBM assumes the existence of adversarial noise $\epsilon_a$ at the starting point of the forward process during training. This means that the starting point of the forward process is $\mathbf{x}_0^a = \mathbf{x}_0 + \epsilon_a$ for each $\mathbf{x}_0$. Thus, according to Eq. (2), the forward process can be represented as $\mathbf{x}_t^a = \sqrt{\bar{\alpha}_t}\mathbf{x}_0^a + \sqrt{1 - \bar{\alpha}_t}\epsilon, \epsilon \sim \mathcal{N}(\mathbf{0}, \mathbf{I}), 0 \leq t \leq T$, where $T$ denotes the actual forward timestep when performing AP. $T$ is typically set to a lower value for AP, *e.g.*, 100 in DiffPure, than that used in generative tasks, *e.g.*, 1,000, to avoid completely corrupting $\mathbf{x}_0$. We discuss how to obtain $\epsilon_a$ in Sec. 4.2.

In the reverse process of ADBM, the objective is to learn a Markov chain $\{\hat{\mathbf{x}}_t\}_{t:T \to 0}$ that can directly transform from the diffused adversarial data distribution (*i.e.*, $\mathbf{x}_T^a$) to the clean data distribution (*i.e.*, $\mathbf{x}_0$), as shown in Figure 1(c). To achieve the goal of purification, we expect that the starting point and the end point of $\hat{\mathbf{x}}_t$ to be $\hat{\mathbf{x}}_T := \mathbf{x}_T^a$ and $\hat{\mathbf{x}}_0 := \mathbf{x}_0$, respectively. Notably, $\hat{\mathbf{x}}_T$ contains adversarial noise, while $\hat{\mathbf{x}}_0$ does not. To explicitly align the trajectory of $\hat{\mathbf{x}}_t$ with the starting and ending points, we introduce a coefficient $k_t$, expecting that $\hat{\mathbf{x}}_t := \mathbf{x}_t^d = \mathbf{x}_t^a - k_t\epsilon_a$ for $0 \leq t \leq T$, with $k_0 = 1$ and $k_T = 0$. With Bayes' rule and the property of Gaussian distribution, we have

$$q(\mathbf{x}_{t-1}^d|\mathbf{x}_t^d, \mathbf{x}_0) = \frac{q(\mathbf{x}_t^d|\mathbf{x}_{t-1}^d, \mathbf{x}_0) \cdot q(\mathbf{x}_{t-1}^d|\mathbf{x}_0)}{q(\mathbf{x}_t^d|\mathbf{x}_0)} \propto \exp\left(-\frac{1}{2}(A(\mathbf{x}_{t-1}^d)^2 + B\mathbf{x}_{t-1}^d + C)\right), \quad (5)$$

where $A$ is a constant independent of $\epsilon_a$, $B$ is the coefficient of $\mathbf{x}_{t-1}^d$ dependent on $\epsilon_a$, and $C$ is a term without $\mathbf{x}_{t-1}^d$. In inference, we expect that $\epsilon_a$ in Eq. (5) can be eliminated since only $\mathbf{x}_0^a$ is given by attackers and $\epsilon_a$ cannot be decoupled from $\mathbf{x}_0^a$ directly. Based on the property of Gaussian distribution, eliminating all terms related to $\epsilon_a$ in $B$ and $C$ in Eq. (5) can be achieved by eliminating all terms related to $\epsilon_a$ in $B$. This yields:

$$\frac{\sqrt{\alpha_t}(\sqrt{\alpha_t}k_{t-1} - k_t)\epsilon_a}{1 - \alpha_t} - \frac{(\sqrt{\bar{\alpha}_{t-1}} - k_{t-1})\epsilon_a}{1 - \bar{\alpha}_{t-1}} = 0. \quad (6)$$

To satisfy $k_0 = 1$, $k_T = 0$, we can derive $k_t$ in Eq. (6) as:

$$k_t = \sqrt{\bar{\alpha}_t} - \frac{\bar{\alpha}_T(1 - \bar{\alpha}_t)}{\sqrt{\bar{\alpha}_t}(1 - \bar{\alpha}_T)}, 0 \leq t \leq T. \quad (7)$$

For detailed derivations, please refer to Appendix C.1.

Following Eq. (3), ADBM uses $\mathbf{x}_\theta(\mathbf{x}_t^d, t) := \frac{1}{\sqrt{\bar{\alpha}_t}}(\mathbf{x}_t^d - \sqrt{1 - \bar{\alpha}_t}\epsilon_\theta(\mathbf{x}_t^d, t))$ to approximate the clean example $\mathbf{x}_0$. Thus the loss of ADBM can be computed by

$$L = \mathbb{E}_{\epsilon,t}\|\mathbf{x}_0 - \mathbf{x}_\theta(\mathbf{x}_t^d, t)\|^2 = \mathbb{E}_{\epsilon,t}\left\|\frac{k_t - \sqrt{\bar{\alpha}_t}}{\sqrt{\bar{\alpha}_t}}\epsilon_a - \frac{\sqrt{1 - \bar{\alpha}_t}}{\sqrt{\bar{\alpha}_t}}(\epsilon - \epsilon_\theta(\mathbf{x}_t^d, t))\right\|^2$$
$$= \mathbb{E}_{\epsilon,t}\left[\frac{1 - \bar{\alpha}_t}{\bar{\alpha}_t}\left\|\frac{\bar{\alpha}_T\sqrt{1 - \bar{\alpha}_t}}{(1 - \bar{\alpha}_T)\sqrt{\bar{\alpha}_t}}\epsilon_a + \epsilon - \epsilon_\theta(\mathbf{x}_t^d, t)\right\|^2\right]. \quad (8)$$

Omitting $\frac{1 - \bar{\alpha}_t}{\sqrt{\bar{\alpha}_t}}$ as in Ho et al. (2020), for a given $\mathbf{x}_0$, the final loss $L_b$ of ADBM is given by:

$$L_b = \mathbb{E}_{\epsilon,t}\left[\left\|\frac{\bar{\alpha}_T\sqrt{1 - \bar{\alpha}_t}}{(1 - \bar{\alpha}_T)\sqrt{\bar{\alpha}_t}}\epsilon_a + \epsilon - \epsilon_\theta(\mathbf{x}_t^d, t)\right\|^2\right], \quad (9)$$

where $\mathbf{x}_t^d = \sqrt{\bar{\alpha}_t}\mathbf{x}_0 + \sqrt{1 - \bar{\alpha}_t}\epsilon + \frac{\bar{\alpha}_T(1 - \bar{\alpha}_t)}{\sqrt{\bar{\alpha}_t}(1 - \bar{\alpha}_T)}\epsilon_a$. The training process of ADBM is detailed in the blue block of Figure 2.

Comparing Eq. (4) and Eq. (9), $L_d$ and $L_b$ are quite similar except that $L_b$ has two additional scaled $\epsilon_a$ in both the input and prediction objective of $\epsilon_\theta$. Thus in practice, the training of ADBM can fine-tune the pre-trained diffusion checkpoint with $L_b$, avoiding training from scratch. As $t$ decreases to 0, the coefficient of $\epsilon_a$ diminishes. This complies with the intuition that $\epsilon_a$ is gradually eliminated.

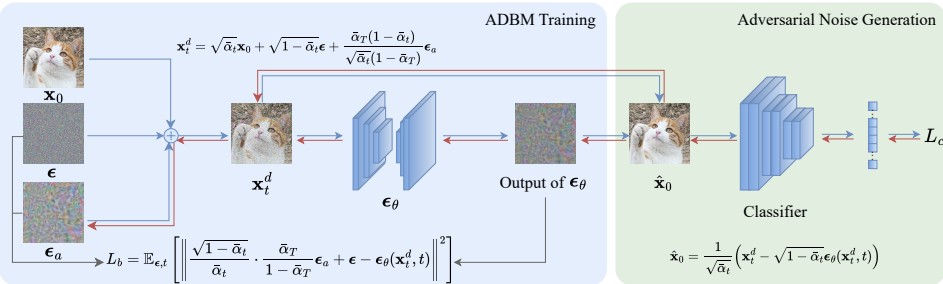

Figure 2: The illustration of ADBM training. The blue block represents the training objective, and the green block represents the extra module for adversarial noise generation. Black arrows denote the computation of $L_b$, blue arrows denote the computation of $L_c$ used for generating adversarial noise, and red arrows denote the direction of the gradient flow when calculating $\boldsymbol{\epsilon}_a$.

## 4.2 Adversarial Noise Generation

We now proceed to the generation of adversarial noise required for ADBM training. A straightforward method for generating adversarial noise can be maximizing the loss $L_b$ of ADBM. However, the idea of deriving adversarial noise directly from the diffusion model itself might not align well with the objectives of the purification task. In this context, the primary goal is to ensure the classifier's accuracy on images after they have been purified.

To better align with this goal, we propose to generate adversarial noise with the help of the classifier. During ADBM training, we input $\hat{\mathbf{x}}_0$ into the classifier, where $\hat{\mathbf{x}}_0 = \frac{1}{\sqrt{\bar{\alpha}_t}}(\mathbf{x}_t^d - \sqrt{1 - \bar{\alpha}_t}\boldsymbol{\epsilon}_\theta(\mathbf{x}_t^d, t))$. The classification loss $L_c$ is then given by $L_c(f_{\theta_c}(\hat{\mathbf{x}}_0), y)$, where $y$ denotes the category label of $\mathbf{x}_0$ and $f_{\theta_c}$ denotes the classifier. Finally, we compute $\frac{\partial L_c}{\partial \boldsymbol{\epsilon}_a}$ to obtain $\boldsymbol{\epsilon}_a$ that maximizes $L_c$. Figure 2 illustrates this process with red arrows indicating the gradient flow direction for maximizing $L_c$. The process of first maximizing $L_c$ then minimizing $L_b$ shares a similar procedure with adversarial training, but we note that: 1) The parameters of the classifier should be kept fixed throughout this process, ensuring that ADBM retains the plug-and-play functionality (without modifying the model to be protected) of AP methods; 2) The introduction of $\boldsymbol{\epsilon}_a$ should align with Eq. (9) to effectively address the trade-off in DiffPure between noise removal and data recovery. In addition, although Lin et al. (2024) designed an impressive AT method named AToP for various AP methods, they claim that AToP is ineffective for diffusion models, highlighting the challenges of enhancing diffusion-based purification models.

Any gradient-based attack method generally can be used for maximizing $L_c$, but in practice, some particular proxy attack should be specified. Based on previous experiences, the PGD attack is a popular proxy that has a good generalization ability against various attacks during inference (Athalye et al., 2018; Croce & Hein, 2020; Madry et al., 2018), and thus we also used such proxy attack in the ADBM training. Our experiments also validate that this proxy can generalize to other attacks such as AutoAttack, C&W (Carlini & Wagner, 2017), and DeepFool (Moosavi-Dezfooli et al., 2016) attacks. Note that during the training of ADBM, both $t$ and $\boldsymbol{\epsilon}$ in Eq. (9) are randomly sampled for each optimization step. This introduces a significant increase in computational cost when maximizing $L_c$ with PGD (EOT should be used). To address this issue, we propose an alternative method: instead of random sampling, we eliminate the randomness within a specific optimization step. In other words, for a given optimization step, we sample $t$ and $\boldsymbol{\epsilon}$ once and then fix their values throughout the entire training step (including generating $\boldsymbol{\epsilon}_a$ using methods like PGD and optimizing $\boldsymbol{\epsilon}_\theta$). The insight behind our approach is that maintaining randomness makes the maximization of $L_c$ via PGD challenging without EOT, and eliminating the randomness within a specific optimization step avoids the time-consuming EOT computation and ensures the calculation of adversarial noise that is truly effective for ADBM training. Note that we only eliminate randomness in the ADBM training. During inference, the randomness is kept, as discussed next.

## 4.3 AP Inference of ADBM

When using ADBM for AP, both the forward and reverse processes remain unchanged from the original pipeline of DiffPure, as shown in Figure 1(a). Any reverse samplers developed for diffusion

models can be directly applied to the AP inference of ADBM without any modification, as ADBM only initiates the reverse process from a different starting point compared to traditional diffusion models. Therefore, to improve the practicality of ADBM, we can leverage fast sampling methods such as DDIM to accelerate the reverse process. As demonstrated in Sec. 3, the DDIM sampler efficiently conducts AP, even with a single reverse step.

## 4.4 THEORETICAL ANALYSIS

We provide two theorems to show the superiority of ADBM for adversarial purification.

**Theorem 1.** *Given an adversarial example $\mathbf{x}_0^a$ and assuming the training loss $L_b \leq \delta$, the distance between the purified example of ADBM and the clean example $\mathbf{x}_0$, denoted as $\|\hat{\mathbf{x}}_0 - \mathbf{x}_0\|$, is bounded by $\delta$ (constant omitted) in expectation when using a one-step DDIM sampler. Specifically, we have $\mathbb{E}_\epsilon \left[ \|\hat{\mathbf{x}}_0 - \mathbf{x}_0\|^2 \right] \leq \frac{(1-\bar{\alpha}_T)T}{\bar{\alpha}_T}\delta$, where $\frac{(1-\bar{\alpha}_T)T}{\bar{\alpha}_T}$ is the constant.*

*Proof.* Please see the full proof in Appendix C.2. □

Theorem 1 implies that if the training loss of ADBM converges to zero, it can perfectly remove adversarial noises by employing a one-step DDIM sampler. While for DiffPure, we cannot derive such strong theoretical guarantee (The bound provided in Theorem 3.2 of Nie et al. (2022) is larger than $\|\epsilon_a\|$ and thus cannot be zero). Moreover, the subsequent theorem demonstrates the superiority of ADBM over DiffPure.

**Theorem 2.** *Denote the probability of reversing the adversarial example to the clean example using ADBM and DiffPure as $P(B)$ and $P(D)$, respectively. Then $P(\cdot) = \int \mathbb{1}_{\{\mathbf{x}_0 \notin \mathbb{D}_a\}} p(\mathbf{x}_0|\hat{\mathbf{x}}_t)\mathrm{d}\mathbf{x}_0$, where $\mathbb{D}_a$ denotes the set of adversarial examples. If the timestep is infinite, the following inequality holds:*

$$P(B) > P(D),$$

*wherein*

$$\text{for } P(B): p(\mathbf{x}_0|\hat{\mathbf{x}}_t) \propto \exp\left(-\frac{\|\mathbf{x}_t^d - \sqrt{\bar{\alpha}_t}\mathbf{x}_0^a\|^2}{2(1-\bar{\alpha}_t)}\right), \tag{10}$$

$$\text{for } P(D): p(\mathbf{x}_0|\hat{\mathbf{x}}_t) \propto \exp\left(-\frac{\|\mathbf{x}_t^a - \sqrt{\bar{\alpha}_t}\mathbf{x}_0\|^2}{2(1-\bar{\alpha}_t)}\right). \tag{11}$$

*Proof.* (sketch) Eq. (10) and Eq. (11) are derived using Bayes' rule, where $p(\mathbf{x}_0|\hat{\mathbf{x}}_t) \propto p(\mathbf{x}_0)p(\hat{\mathbf{x}}_t|\mathbf{x}_0)$. From the perspective of SDE, if timestep is infinite, $\{\mathbf{x}_t\}_{t:0\to T}$ and $\{\hat{\mathbf{x}}_t\}_{t:T\to 0}$ follow the same distribution (Song et al., 2021). And given that $k_t < \sqrt{\bar{\alpha}_t}$ for any $1 \leq t \leq T$, the inequality $P(B) > P(D)$ always holds. Please see the full proof in Appendix C.3. □

Theorem 2 indicates that with infinite reverse timesteps, adversarial examples purified with ADBM are more likely to align with the clean data distribution than those with DiffPure.

## 5 EXPERIMENTS

### 5.1 EXPERIMENTAL SETTINGS

**Datasets and network architectures.** We conducted comprehensive experiments on popular datasets, including SVHN (Netzer et al., 2011), CIFAR-10 (Krizhevsky et al., 2009), and Tiny-ImageNet (Le & Yang, 2015), together with a large-scale dataset ImageNet-100 [2] All these datasets consist of RGB images, whose resolution is $32 \times 32$ for SVHN and CIFAR-10, $64 \times 64$ for Tiny-ImageNet, and $224 \times 224$ for ImageNet-100. We adopted the widely used WideResNet-28-10 (WRN-28-10), WRN-70-16, WRN-28-10, and ResNet-50 (He et al., 2016) architectures as classifiers on SVHN, CIFAR-10, Tiny-ImageNet, and ImageNet-100, respectively. As for the diffusion

---

[2] https://www.kaggle.com/datasets/ambityga/imagenet100. We cannot afford to conduct experiments on the full ImageNet-1K (Russakovsky et al., 2015). As an alternative, we used ImageNet-100, a curated subset of ImageNet-1K featuring 100 categories with the original resolution of ImageNet-1K.

models, we employed the UNet architecture (Ronneberger et al., 2015) improved by Song et al. (2021), specifically, the `DDPM++ continuous` variant. Pre-trained diffusion checkpoints are required for DiffPure. We directly used the checkpoint provided by Song et al. (2021) for CIFAR-10 and we used their code to train the checkpoints for other datasets. These trained checkpoints were used in DiffPure and served as baselines for ADBM.

**Fine-tuning settings of ADBM.** The adversarial noise was computed in the popular norm-ball setting $\|\epsilon_a\|_\infty \leq 8/255$. When computing $\epsilon_a$, we used PGD with three iteration steps and a step size of $8/255$. Other settings followed the standard configuration used in Song et al. (2021). The fine-tuning steps were set to 30K, which is about 1/10 the training steps of the original diffusion models. In each fine-tuning step, the value of $T$ in Eq. (9) was uniformly sampled from 100 to 200. Note that when fine-tuning the diffusion models, the parameters of the classifier were kept frozen. Additional settings are provided in Appendix D.1.

**Defense configurations of ADBM.** Unless otherwise specified, the forward diffusion steps were set to be 100 for SVHN and CIFAR-10 and 150 for Tiny-ImageNet and ImageNet-100, respectively. The reverse sampling steps were set to be five. The reverse process used a DDIM sampler. These configurations were also applied to DiffPure for a fair comparison.

Table 3: Accuracies (%) of methods under different adaptive attack threats on CIFAR-10. *Average* denotes the average accuracies under three attack threats. *Vanilla* denotes the vanilla model without any defense mechanism. The best results in each column for robust accuracy are highlighted.

| Architecture | Method | Type | Clean Acc | Robust Acc | | | |
|---|---|---|---|---|---|---|---|
| | | | | $l_\infty$ norm | $l_1$ norm | $l_2$ norm | Average |
| WRN-70-16 | Vanilla | - | 97.02 | 0.00 | 0.00 | 0.00 | 0.00 |
| WRN-70-16 | (Gowal et al., 2020) | AT | 91.10 | 65.92 | 8.26 | 27.56 | 33.91 |
| WRN-70-16 | (Rebuffi et al., 2021) | | 88.54 | 64.26 | 12.06 | 32.29 | 36.20 |
| WRN-70-16 | Augment w/ Diff (Gowal et al., 2021) | | 88.74 | 66.18 | 9.76 | 28.73 | 34.89 |
| WRN-70-16 | Augment w/ Diff (Wang et al., 2023) | | 93.25 | **70.72** | 8.48 | 28.98 | 36.06 |
| MLP+WRN-28-10 | (Shi et al., 2021) | AP | 91.89 | 4.56 | 8.68 | 7.25 | 6.83 |
| UNet+WRN-70-16 | (Yoon et al., 2021) | | 87.93 | 37.65 | 36.87 | 57.81 | 44.11 |
| UNet+WRN-70-16 | DiffPure+Guide (Wang et al., 2022) | | 93.16 | 22.07 | 28.71 | 35.74 | 28.84 |
| UNet+WRN-70-16 | Diff+ScoreOpt (Zhang et al., 2024) | | 91.41 | 13.28 | 10.94 | 28.91 | 17.71 |
| UNet+WRN-70-16 | DiffPure+Langevin (Zhang et al., 2023) | | 92.18 | 43.75 | 39.84 | 55.47 | 46.35 |
| UNet+WRN-70-16 | DiffPure (Nie et al., 2022) | | $92.5 \pm 0.5$ | $42.2 \pm 2.1$ | $44.3 \pm 1.3$ | $60.8 \pm 2.3$ | $49.1 \pm 1.7$ |
| UNet+WRN-70-16 | ADBM (Ours) | | $91.9 \pm 0.8$ | $47.7 \pm 2.2$ | $\mathbf{49.6 \pm 2.2}$ | $\mathbf{63.3 \pm 1.9}$ | $\mathbf{53.5 \pm 2.1}$ |

## 5.2 ROBUSTNESS AGAINST WHITE-BOX ADAPTIVE ATTACKS

We first evaluate ADBM against the reliable while-box adaptive attacks (Athalye et al., 2018) to show the worst-case adversarial robustness where the attacker has complete knowledge. Note that we expect a defense method to be robust not only on seen threats but also on unseen attack threats. Thus, unless otherwise specified, we evaluated the models on three attack threats: $l_\infty$, $l_1$, and $l_2$, with the bounds $\epsilon_\infty = 8/255$, $\epsilon_1 = 12$, $\epsilon_2 = 1$, respectively. Here $l_\infty$ attack is considered the seen threat as ADBM was trained with $l_\infty$ adversarial noise, while $l_1$ and $l_2$ attacks can be regarded as unseen threats.

We compared ADBM with SOTA AT and AP methods. All AT models were trained also with $l_\infty$ adversarial examples only, ensuring $l_1$ and $l_2$ threats were unseen for these models. We used AutoAttack (Croce & Hein, 2020), which is recognized as the strongest attacks for AT methods, to implement the adversarial threats for AT methods, while we used the attack practice described in Sec. 3 (PGD with 200 iteration steps and 20 EOT samples) to implement the adversarial threats for AP methods. Additional configurations of these attacks can be found in Appendix D.2. Note that this evaluation comparison is justified since Lee & Kim (2023) observed that AutoAttack with EOT yielded inferior attack performance compared to PGD with EOT for stochastic pre-processing defenses. Our experiments in Appendix E.1 further confirmed that PGD with EOT is the best practice than other attacks such as AutoAttack, C&W, and DeepFool with EOT.

The accuracies under the reliable white-box adaptive attacks on CIFAR-10 are shown in Tab. 3. In this evaluation, we compared ADBM with several competitive methods. Both Gowal et al. (2021) and Wang et al. (2023) performed AT with about 50M generated images by diffusion models. Addi-

Table 4: Accuracies (%) of methods under three query-based attacks and the transfer-based attack on SVHN. *Average* denotes the average accuracies under four attacks. All attacks are performed with the $l_\infty$ bound $8/255$.

| Architecture | Method | Type | Clean Acc | Robust Acc | | | | |
|---|---|---|---|---|---|---|---|---|
| | | | | RayS | Square | SPSA | Transfer | Average |
| WRN-28-10 | Vanilla | - | 98.11 | 16.89 | 9.08 | 13.48 | - | - |
| WRN-28-10 | (Rade et al., 2022) | | 94.46 | 68.90 | 58.96 | 74.22 | 87.60 | 76.83 |
| WRN-28-10 | (Yang et al., 2023) | AT | 93.00 | 62.30 | 60.49 | 71.78 | 87.60 | 75.53 |
| WRN-28-10 | (Wang et al., 2023) | | 95.56 | 75.16 | 67.23 | 80.39 | 88.57 | 81.38 |
| UNet+WRN-28-10 | DiffPure (Nie et al., 2022) | AP | 93.93 | 92.97 | 92.15 | 92.19 | 91.49 | 92.20 |
| UNet+WRN-28-10 | ADBM (Ours) | | 93.49 | **93.16** | **93.32** | **93.49** | **92.88** | **93.21** |

tionally, we compared ADBM with several recent AP methods using generative models, especially diffusion models.[3] By inspecting these results, we can see that despite training with millions of examples, AT methods still exhibited limited robustness against unseen $l_1$ and $l_2$ attacks. In contrast, ADBM demonstrates a strong defense against $l_1$ and $l_2$ attacks. Moreover, attempts to enhance the performance of DiffPure by introducing input guidance (DiffPure+Guide, Wang et al. (2022)) or applying Langevin dynamics (DiffPure+Langevin, Zhang et al. (2023)) occasionally resulted in detrimental effects when assessed through reliable adaptive attack evaluations. Notably, ADBM outperformed DiffPure by achieving an average robustness gain of 4.4% on CIFAR-10, while the clean accuracies kept comparable. Similar outcomes were observed for SVHN, Tiny-ImageNet, and ImageNet-100, detailed in Appendix E.2, reinforcing similar findings from the CIFAR-10 analyses.

## 5.3 ROBUSTNESS AGAINST BLACK-BOX ATTACKS

We then considered the more realistic black-box attacks, where the attacker has no knowledge about the defense mechanism, *i.e.*, the purification model, and cannot access the gradients of models. Instead, the attacker can only query the model's output with query-based attacks or use substitute models with transfer-based attacks.

On the seen threat (*i.e.*, $l_\infty$ threat), we conducted three query-based attacks: RayS (Chen & Gu, 2020), Square (Andriushchenko et al., 2020), and SPSA (Uesato et al., 2018) attacks. Square and RayS are efficient search-based attack methods, while SPSA approximates gradients in a black-box manner. For Square and RayS, we utilized 5,000 search steps. For SPSA, we set $\sigma$ to 0.001, the number of random samples in each step to 128, and the iteration step to 40. In addition, We implemented a transfer-based attack by generating adversarial examples via the *vanilla* models. The results under black-box attacks on SVHN and CIFAR-10 are shown in Tab. 4 and Appendix E.3, respectively. We can see that these black-box attacks achieved excellent attack performance on the vanilla model. For the SOTA AT method on SVHN, these attacks lowered the average accuracy to 81.38%. But surprisingly, all these black-box attacks can hardly lower the accuracies of ADBM. Thus, we can conclude that besides the promising results on white-box attacks, under the realistic black-box attacks, ADBM has advantages over AT models even on the seen threat.

We further conducted experiments on more unseen threats beyond norm-bounded attacks, including patch-like attacks (Gao et al., 2020; Yuan et al., 2022) and recent diffusion-based attacks (Chen et al., 2023b; Xue et al., 2023), to evaluate the generalization ability of ADBM. The results shown in Appendix E.4 indicate that ADBM demonstrates better generalization ability than DiffPure and AT methods against these black-box unseen threats.

## 5.4 ABLATION STUDY

The ablation studies were performed on CIFAR-10. The evaluation followed the setting in Sec. 5.2.

**Reverse sampling steps.** We first investigated the influence of reverse sampling steps on the adversarial robustness of ADBM. The number of reverse steps is proportional to the inference cost. We used five reverse steps in the main experiments. Here we evaluated the robustness of ADBM with

---

[3]RDC (Chen et al., 2023a) was not compared as it required thousands of times the inference cost of DiffPure so that we cannot afford to conduct experiments in our setting. Instead, we discuss it in Appendix A.2.

Table 5: Accuracies (%) of methods under different adaptive attack threats on CIFAR-10. The same conventions are used as in Tab. A4.

| Method | Reverse Step | Clean | Robust Acc | | | |
|--------|--------------|-------|-----------|----|----|---------|
| | | | $l_\infty$ | $l_1$ | $l_2$ | Average |
| DiffPure | 5 | 92.5 | 42.2 | 44.3 | 60.8 | 49.1 |
| | 2 | 92.3 | 42.7 | 44.5 | 60.9 | 49.4 |
| | 1 | 92.3 | 43.0 | 45.2 | 59.6 | 49.3 |
| ADBM | 5 | 91.9 | **47.7** | 49.6 | **63.3** | **53.5** |
| | 2 | 91.9 | **47.7** | 49.2 | 63.2 | 53.3 |
| | 1 | 91.5 | 45.7 | **50.7** | 61.9 | 52.8 |

Table 6: Accuracies (%) of ADBM with various adversarial noise generating modes on CIFAR-10. *Cls* indicates whether or not to employ the classifier when generating adversarial noise for training.

| Fixing $t$ | Fixing $\mathbf{x}$ | Cls | $l_\infty$ | $l_1$ | $l_2$ | Average |
|-----------|---------------------|-----|-----------|-------|-------|---------|
| ✓ | ✓ | | 44.0 | 44.9 | 61.4 | 50.1 |
| | ✓ | ✓ | 44.3 | 46.2 | 60.2 | 50.2 |
| ✓ | | ✓ | 43.7 | 45.6 | 60.3 | 50.2 |
| ✓ | ✓ | ✓ | **47.7** | **49.6** | **63.3** | **53.5** |

fewer steps to investigate whether the number of steps can be further reduced. The results in Tab. 5 show ADBM was more robust than DiffPure regardless of reverse steps. Notably, even with just one reverse step, ADBM maintained its good robustness. We discuss the detailed inference cost related to this further in Sec. 6.

**Adversarial noise generation modes.** We then analyzed the impact of different adversarial noise generation modes for ADBM. As discussed in Sec. 4.2, adversarial noise used for training ADBM can be generated by various modes. We analyzed the contributions of our three key design choices in the generation modes: using the classifier, fixing $t$, and fixing $\mathbf{x}$. The results shown in Tab. 6 clearly demonstrate that all of these designs are essential for the success of ADBM. Removing any of these designs hampers the proper computation of adversarial noise for ADBM training.

**Effectiveness on new classifiers.** To assess the effectiveness of ADBM on new classifiers, we conducted a study to investigate its transferability. Specifically, we utilized the fine-tuned ADBM checkpoint, trained with adversarial noise from a WRN-70-16 classifier, as the pre-processor for a WRN-28-10 model and a vision transformer model (Dosovitskiy et al., 2021) directly, denoted as ADBM (Transfer). The results shown in Appendix E.5 demonstrate that ADBM (Transfer) achieved robust accuracies comparable to ADBM directly trained with corresponding classifiers. This finding highlights the practicality of ADBM, as the fine-tuned ADBM model on a specific classifier can potentially be directly applied to a new classifier without requiring retraining. We guess this may be attributed to the transferability of adversarial noise (Dong et al., 2018).

## 6 CONCLUSION AND DISCUSSION

In this work, we introduce ADBM, a cutting-edge method for diffusion-based adversarial purification. Theoretical analysis supports the superiority of ADBM in enhancing adversarial robustness. With extensive experiments, we demonstrated the effectiveness of ADBM across various scenarios using reliable adaptive attacks. Notably, ADBM demonstrates significant improvements over prevalent AT methods against unseen threats or black-box attacks. In the era of foundation models (Bommasani et al., 2021), training foundation models with AT becomes increasingly challenging due to high computational costs. ADBM, on the other hand, provides a promising alternative as a plug-and-play component that can enhance the robustness of existing foundation models without the burdensome of retraining. In addition, our discussion on the acceleration of diffusion-based purification methods may unlock the potential of diffusion-based purification methods in various real-time applications (see Appendix F). The social impact of this work is further discussed in Appendix G.

**Limitation.** 1) While ADBM requires quite few reverse steps, it does introduce additional computational parameters due to its reliance on diffusion models compared with AT methods. The current UNet architectures, tailored primarily for generative purposes, are a bit large (see Appendix F). Exploring the potential of downsizing these architectures for AP remains an open area. 2) While ADBM requires only about 1/10 training steps of original diffusion models, it does introduce additional fine-tuning cost compared with using the pre-trained diffusion models directly (DiffPure). But we note that whether the slight extra training cost is worthwhile depends on the specific problems being addressed. In many security-critical cases, where the training cost is not the primary concern, ADBM offers an alternative approach to further enhance robustness with affordable training costs.

ACKNOWLEDGEMENT

This work was supported by the National Natural Science Foundation of China (No. U2341228).

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

## A   DISCUSSIONS ON DIFFUSION BRIDGES AND DIFFUSION CLASSIFIERS

### A.1   DIFFUSION BRIDGE MODELS

Bortoli et al. (2021) is fundamentally unsuitable for the AP task because it bridges the data distribution with a simple prior distribution, while Zhou et al. (2024); Bortoli et al. (2021); Liu et al. (2023b) tend to a training collapse for the AP task. Here we provide the theoretical explanation: Adversarial examples typically reside close to clean examples, particularly in the commonly analyzed $l_\infty$-bounded threat settings. This proximity poses a significant challenge to applying these bridge models (Zhou et al., 2024; Bortoli et al., 2021; Liu et al., 2023b) for AP, as these bridge models' merely predicting the input adversarial image could result in extremely small training loss, finally incurring a degenerated solution and losing the capability to purify the adversarial perturbations.

Different from these bridge models, ADBM first maps the adversarial data distribution to a diffused (Gaussian) adversarial data distribution and then creates a bridge from the diffused adversarial data distribution to the clean data distribution. This is distinct from existing diffusion bridges, effectively solving the problems that previous diffusion bridges face in purification. Our theoretical analysis in Sec. 4.4 consolidates that ADBM guarantees superior performances for AP.

### A.2   ROBUST DIFFUSION CLASSIFIERS

Chen et al. (2023a) show that a single diffusion model can be transformed into an adversarially robust diffusion classifier (RDC) using Bayes' rule. Although RDC does not require an additional classifier, ADBM has two unique and significant advantages over RDC:

1) ADBM is an adversarial purification method, which can function as a preprocessing module in a plug-and-play manner without retraining the classifiers. This is a quite flexible and useful way, especially in the era of foundation models. For example, we may use ADBM as a preprocessing step to purify the visual inputs of a large vision-language model (VLM), without retraining the VLM with adversarial training (huge computational costs are saved). However, RDC does not follow this paradigm, as it is a robust classifier instead of a robust pre-processor. RDC is limited to generalize to tasks beyond robust classification, hindering its practical usefulness. Overall, ADBM offers a distinct advantage over RDC by decoupling preprocessing and the specific classification task, allowing each to be optimized independently.

2) While diffusion-based purification requires slightly more parameters (an additional classifier), ADBM requires significantly lower inference costs than RDC. Our following experiments indicate that, even with one reverse step, ADBM can maintain substantial robustness under reliable evaluation. In contrast, RDC requires N + T reverse steps, where T is the diffusion steps (*e.g.*, 1000 in DDPM) and N is the number of likelihood maximization steps. The inference cost is proportional to the reverse steps, and thus the inference cost of RDC is several hundred times that of ADBM.

## B   EVALUATIONS ON DIFFPURE

### B.1   PREVIOUS EVALUATION METHODS ON DIFFPURE

As the original implementation of DiffPure needs dozens of prediction steps in the reverse process, it is challenging to compute the full gradient of the whole purification process due to memory constraints. To circumvent computing the full gradient, Nie et al. (2022) originally employed the *adjoint* method (along with Expectation over Transformation (EOT) (Athalye et al., 2018)) to compute an approximate gradient. But recent works have identified it to be a flawed adaptive attack (Lee & Kim, 2023; Chen et al., 2023a). As improved adaptive mechanisms, Lee & Kim (2023) proposed a *surrogate* attack for DiffPure (to approximate the gradient of the iterative reverse procedure better) and Chen et al. (2023a) utilized the gradient-checkpointing technique (Chen et al., 2016) to trade time for memory space and compute the full gradient directly. Despite previous efforts, we find that the adaptive evaluations for DiffPure remained insufficient. Specifically, the surrogate attack failed to compute the full gradient, and the gradient-checkpointing attack in Chen et al. (2023a) employed insufficient iteration steps or EOT samples, an issue underscored by Gao et al. (2022) which highlights the importance of adequate steps and samples for evaluating stochastic pre-processing defenses like DiffPure.

Table A1: Accuracies of DiffPure on CIFAR-10 with reverse step 5, varying forward steps, and using DDIM sampler. The robust accuracy was evaluated under an $l_\infty$ bound of $\epsilon_\infty = 8/255$.

| Forward Step | 100 | 110 | 120 | 130 | 140 | 150 |
|---|---|---|---|---|---|---|
| Clean Acc | **92.38** | 91.21 | 91.01 | 89.84 | 89.45 | 87.89 |
| Robust Acc | 42.16 | 43.75 | 44.53 | 47.85 | 48.43 | **49.02** |

### B.2 THE COST OF OUR EVALUATION

In our evaluation, we used the full gradient of the whole reverse process and set the PGD iteration steps to 200, with 20 EOT samples for each iteration. The cost of the evaluation is quite high, especially in the context of the high reverse steps of the original DiffPure. To give a concrete example, for a single input image, the noise prediction model (*i.e.*, $\epsilon_\theta$) in the original DiffPure implementation with 100 reverse steps needs to be queried a total of $400,000$ times.

However, we think that such high efforts are worthwhile as an unreliable evaluation could create a false sense of security on defenses. Historical evidence has shown that many defenses initially considered robust, were subsequently breached by more sophisticated and dedicated attacks (Athalye et al., 2018; Tramèr et al., 2020). Our aim is to prevent a similar outcome for diffusion-based purification, advocating for the employment of a meticulous and reliable attack evaluation methodology, regardless of the expense involved.

On the other hand, with our reliable evaluation, we have investigated the influence of forward steps and reverse steps on the robustness of diffusion-based purification. Our following results in Tab. 2 indicate that reverse steps will not significantly influence the robustness of DiffPure under the reliable evaluation. This suggests that diffusion-based purification techniques might not benefit from increasing the number of reverse steps to complicate the attack process, as such strategies could finally be neutralized by high-cost attacks similar to the one we have implemented. And when the reverse steps are reduced (*e.g.*, five steps in our main experiments), the attack cost of our evaluation method is comparable to the most widely used AutoAttack benchmark (Croce & Hein, 2020).

### B.3 ADDITIONAL INVESTIGATION ON DIFFPURE WITH OUR EVALUATION

As shown in Tab. 2, we find that the reverse step does not significantly affect the robustness when using the DDIM sampler. Considering the computational cost of attacks, we fixed the reverse step to 5 with the DDIM sampler and focused on investigating the influence of forward steps on robustness. The results are shown in Tab. A1. We observed a continuous decrease in clean accuracy and a continuous increase in robustness as the number of forward steps increased. This can be attributed to the introduction of more noise during the forward process.

## C DERIVATION OF EQUATIONS AND PROOFS OF THEOREMS

### C.1 THE DERIVATION OF EQ. (7)

*Proof.* Following the Bayes' rule, we have

$$q(\mathbf{x}_{t-1}^d|\mathbf{x}_t^d, \mathbf{x}_0) = \frac{q(\mathbf{x}_t^d|\mathbf{x}_{t-1}^d, \mathbf{x}_0) \cdot q(\mathbf{x}_{t-1}^d|\mathbf{x}_0)}{q(\mathbf{x}_t^d|\mathbf{x}_0)} \tag{12}$$

Since $\mathbf{x}_t^d = \mathbf{x}_t^a - k_t\epsilon_a$ and $\mathbf{x}_t^a = \sqrt{\bar{\alpha}_t}\mathbf{x}_0^a + \sqrt{1-\bar{\alpha}_t}\epsilon$, there is

$$\mathbf{x}_t^d = \sqrt{\bar{\alpha}_t}\mathbf{x}_0 + \sqrt{1-\bar{\alpha}_t}\epsilon + (\sqrt{\bar{\alpha}_t} - k_t)\epsilon_a. \tag{13}$$

Based on the property of Gaussian distribution, $p(\mathbf{x}_{t-1}^d|\mathbf{x}_t^d, \mathbf{x}_0)$ also must be Gaussian distribution, thus,

$$
\begin{aligned}
q(\mathbf{x}_{t-1}^d|\mathbf{x}_t^d, \mathbf{x}_0) &= \frac{q(\mathbf{x}_t^d|\mathbf{x}_{t-1}^d, \mathbf{x}_0) \cdot q(\mathbf{x}_{t-1}^d|\mathbf{x}_0)}{q(\mathbf{x}_t^d|\mathbf{x}_0)} \propto \exp(-\frac{1}{2}(\frac{(\mathbf{x}_t^d - \sqrt{\alpha_t}\mathbf{x}_{t-1}^d - (\sqrt{\alpha_t}k_{t-1} - k_t)\boldsymbol{\epsilon}_a)^2}{1 - \alpha_t} \\
&\quad + \frac{(\mathbf{x}_{t-1}^d - \sqrt{\bar{\alpha}_{t-1}}\mathbf{x}_0 - (\sqrt{\bar{\alpha}_{t-1}} - k_{t-1})\boldsymbol{\epsilon}_a)^2}{1 - \bar{\alpha}_{t-1}} - \frac{(\mathbf{x}_t^d - \sqrt{\bar{\alpha}_t}\mathbf{x}_0 - (\sqrt{\bar{\alpha}_t} - k_t)\boldsymbol{\epsilon}_a)^2}{1 - \bar{\alpha}_t})) \\
&= \exp\left(-\frac{1}{2}(A(\mathbf{x}_{t-1}^d)^2 + B\mathbf{x}_{t-1}^d + C(\mathbf{x}_0, \boldsymbol{\epsilon}_a, \mathbf{x}_t^d))\right),
\end{aligned}
\tag{14}
$$

where

$$
\begin{aligned}
A &= \frac{\alpha_t}{1 - \alpha_t} + \frac{1}{1 - \bar{\alpha}_{t-1}} = \frac{1 - \bar{\alpha}_t}{(1 - \alpha_t)(1 - \bar{\alpha}_{t-1})}, \\
B &= -2\sqrt{\alpha_t} \cdot \frac{\mathbf{x}_t^d - (\sqrt{\alpha_t}k_{t-1} - k_t)\boldsymbol{\epsilon}_a}{1 - \alpha_t} - 2\frac{\sqrt{\bar{\alpha}_{t-1}}\mathbf{x}_0 + (\sqrt{\bar{\alpha}_{t-1}} - k_{t-1})\boldsymbol{\epsilon}_a}{1 - \bar{\alpha}_{t-1}}.
\end{aligned}
\tag{15}
$$

In inference, we expect that $\boldsymbol{\epsilon}_a$ in Eq. (14) can be eliminated since only $\mathbf{x}_0^a$ is given by attackers and $\boldsymbol{\epsilon}_a$ cannot be decoupled from $\mathbf{x}_0^a$ directly. Based on the property of Gaussian distribution, eliminating all terms related to $\boldsymbol{\epsilon}_a$ in $B$ and $C$ in Eq. (5) can be achieved by eliminating all terms related to $\boldsymbol{\epsilon}_a$ in $B$. This yields:

$$
\frac{\sqrt{\alpha_t}(\sqrt{\alpha_t}k_{t-1} - k_t)\boldsymbol{\epsilon}_a}{1 - \alpha_t} - \frac{(\sqrt{\bar{\alpha}_{t-1}} - k_{t-1})\boldsymbol{\epsilon}_a}{1 - \bar{\alpha}_{t-1}} = 0.
\tag{16}
$$

Although Eq. (16) cannot be directly solved, we can derive its recurrent form. Let us set $k_t = \sqrt{\bar{\alpha}_t}\gamma_t$, where $0 \le t \le T$ and $\gamma_0 = 1, \gamma_T = 0$. The Eq. (16) can deduce to

$$
\frac{\sqrt{\alpha_t}}{1 - \alpha_t}(\sqrt{\bar{\alpha}_t}\gamma_{t-1} - \sqrt{\bar{\alpha}_t}\gamma_t) = \frac{\sqrt{\bar{\alpha}_{t-1}} - \sqrt{\bar{\alpha}_{t-1}}\gamma_{t-1}}{1 - \bar{\alpha}_{t-1}}.
\tag{17}
$$

Then group the items according to timestep ($t$ and $t - 1$):

$$
\left(\frac{\alpha_t}{1 - \alpha_t} + \frac{1}{1 - \bar{\alpha}_{t-1}}\right)\gamma_{t-1} = \frac{1}{1 - \bar{\alpha}_{t-1}} + \frac{\alpha_t}{1 - \alpha_t}\gamma_t.
\tag{18}
$$

Now we have a recurrent equation about $\gamma_t$. The equivalence holds for all $0 < t \le T$. Therefore, by elucidating the relationship between $\gamma_t$ and the initial value $\gamma_1$, we can deduce an expression for each $\gamma_t$. If we reorganize items in Eq. (18), it yields a recurrent equation:

$$
\begin{aligned}
\gamma_t - 1 &= \frac{1 - \bar{\alpha}_t}{\alpha_t - \bar{\alpha}_t}(\gamma_{t-1} - 1) \\
&= \prod_{i=2}^{t} \frac{1 - \bar{\alpha}_i}{\alpha_i - \bar{\alpha}_i}(\gamma_1 - 1) \\
&= \frac{\bar{\alpha}_1(1 - \bar{\alpha}_t)}{\bar{\alpha}_t(1 - \bar{\alpha}_1)}(\gamma_1 - 1).
\end{aligned}
\tag{19}
$$

Since $\gamma_T = 0$, we have the expression of $\gamma_1$ when $t = T$:

$$
\begin{aligned}
\gamma_T - 1 &= \frac{\bar{\alpha}_1(1 - \bar{\alpha}_T)}{\bar{\alpha}_T(1 - \bar{\alpha}_1)}(\gamma_1 - 1), \\
\gamma_1 &= 1 - \frac{\bar{\alpha}_T(1 - \bar{\alpha}_1)}{\bar{\alpha}_1(1 - \bar{\alpha}_T)}
\end{aligned}
\tag{20}
$$

and $\gamma_t$ generated from $\gamma_1$ is

$$
\gamma_t = 1 - \frac{\bar{\alpha}_T(1 - \bar{\alpha}_t)}{\bar{\alpha}_t(1 - \bar{\alpha}_T)}.
\tag{21}
$$

Recalling that $k_t = \sqrt{\bar{\alpha}_t}\gamma_t$, we thus have

$$
k_t = \sqrt{\bar{\alpha}_t}\gamma_t = \sqrt{\bar{\alpha}_t} - \frac{\bar{\alpha}_T(1 - \bar{\alpha}_t)}{\sqrt{\bar{\alpha}_t}(1 - \bar{\alpha}_T)}, \quad 0 \le t \le T.
\tag{22}
$$

$\square$

## C.2 THE PROOF OF THEOREM 1

**Theorem 1.** *Given an adversarial example $\mathbf{x}_0^a$ and assuming the training loss $L_b \leq \delta$, the distance between the purified example of ADBM and the clean example $\mathbf{x}_0$, denoted as $\|\hat{\mathbf{x}}_0 - \mathbf{x}_0\|$, is bounded by $\delta$ in expectation (constant omitted) when using a one-step DDIM sampler. Specifically, we have $\mathbb{E}_{\boldsymbol{\epsilon}}[\|\hat{\mathbf{x}}_0 - \mathbf{x}_0\|^2] \leq \frac{(1-\bar{\alpha}_T)T}{\bar{\alpha}_T}\delta$, where $\frac{(1-\bar{\alpha}_T)T}{\bar{\alpha}_T}$ is the constant.*

*Proof.* This inequality holds when we use the DDIM reverse sampler and set the reverse step $s = 1$. According to Song et al. (2020), the reverse process of DDIM is

$$\hat{\mathbf{x}}_{\tau_{i-1}} = \sqrt{\bar{\alpha}_{\tau_{i-1}}}\left(\frac{\hat{\mathbf{x}}_{\tau_i} - \sqrt{1-\bar{\alpha}_{\tau_i}}\boldsymbol{\epsilon}_\theta(\hat{\mathbf{x}}_{\tau_i}, \tau_i)}{\sqrt{\bar{\alpha}_{\tau_i}}}\right) + \sqrt{1-\bar{\alpha}_{\tau_{i-1}}}\boldsymbol{\epsilon}_\theta(\hat{\mathbf{x}}_{\tau_i}, \tau_i). \tag{23}$$

In the discrete case (Ho et al., 2020), $\{\tau_0, \ldots, \tau_s\}$ is a linearly increasing sub-sequence of $\{0, \ldots, T\}$, $\tau_0 = 0, \tau_s = T$. In the continuous case (Song et al., 2021), $\{\tau_0, \ldots, \tau_s\}$ is a linearly increasing sequence in $[0, T] \subset [0.0, 1.0]$, $\tau_0 = 0, 0 \leq \tau_s = T \leq 1$. $T$ is determined in the forward process. Setting the number of reverse steps $s = 1$, then $\tau_s = T, \tau_{s-1} = \tau_0 = 0$, and it yields:

$$\hat{\mathbf{x}}_{\tau_{s-1}} = \sqrt{\bar{\alpha}_{\tau_{s-1}}}\left(\frac{\hat{\mathbf{x}}_{\tau_s} - \sqrt{1-\bar{\alpha}_{\tau_s}}\boldsymbol{\epsilon}_\theta(\hat{\mathbf{x}}_{\tau_s}, \tau_s)}{\sqrt{\bar{\alpha}_{\tau_s}}}\right) + \sqrt{1-\bar{\alpha}_{\tau_{s-1}}}\boldsymbol{\epsilon}_\theta(\hat{\mathbf{x}}_{\tau_s}, \tau_s), \tag{24}$$

since $\hat{\mathbf{x}}_{\tau_s} = \hat{\mathbf{x}}_T = \mathbf{x}_T^a$, we have

$$\hat{\mathbf{x}}_0 = \sqrt{\bar{\alpha}_0}\left(\frac{\mathbf{x}_T^a - \sqrt{1-\bar{\alpha}_T}\boldsymbol{\epsilon}_\theta(\mathbf{x}_T^a, T)}{\sqrt{\bar{\alpha}_T}}\right) + \sqrt{1-\bar{\alpha}_0}\boldsymbol{\epsilon}_\theta(\mathbf{x}_T^a, T), \tag{25}$$

where since we use one-step reverse process, we reuse the notation of the final point of reverse process $\hat{\mathbf{x}}_0$ to represent the prediction of $\mathbf{x}_0$ from $\mathbf{x}_T^a$. Since $\sqrt{\bar{\alpha}_0} = 1$, $\mathbf{x}_T^a = \sqrt{\bar{\alpha}_T}\mathbf{x}_0^a + \sqrt{1-\bar{\alpha}_T}\boldsymbol{\epsilon}$, $\mathbf{x}_0^a = \mathbf{x}_0 + \boldsymbol{\epsilon}_a$, Eq. (25) can be written as

$$\begin{aligned}
\hat{\mathbf{x}}_0 &= \mathbf{x}_0^a + \frac{\sqrt{1-\bar{\alpha}_T}}{\sqrt{\bar{\alpha}_T}}(\boldsymbol{\epsilon} - \boldsymbol{\epsilon}_\theta(\mathbf{x}_T^a, T)) \\
&= \mathbf{x}_0 + \frac{\sqrt{1-\bar{\alpha}_T}}{\sqrt{\bar{\alpha}_T}}\left(\frac{\sqrt{\bar{\alpha}_T}}{\sqrt{1-\bar{\alpha}_T}}\boldsymbol{\epsilon}_a + \boldsymbol{\epsilon} - \boldsymbol{\epsilon}_\theta(\mathbf{x}_T^a, T)\right).
\end{aligned} \tag{26}$$

Therefore, the distance between $\hat{\mathbf{x}}_0$ and $\mathbf{x}_0$ is

$$\begin{aligned}
\|\hat{\mathbf{x}}_0 - \mathbf{x}_0\| &= \frac{\sqrt{1-\bar{\alpha}_T}}{\sqrt{\bar{\alpha}_T}}\left\|\frac{\sqrt{\bar{\alpha}_T}}{\sqrt{1-\bar{\alpha}_T}}\boldsymbol{\epsilon}_a + \boldsymbol{\epsilon} - \boldsymbol{\epsilon}_\theta(\mathbf{x}_T^a, T)\right\| \\
&= \frac{\sqrt{1-\bar{\alpha}_T}}{\sqrt{\bar{\alpha}_T}}\left\|\frac{\sqrt{\bar{\alpha}_T}}{\sqrt{1-\bar{\alpha}_T}}\boldsymbol{\epsilon}_a + \boldsymbol{\epsilon} - \boldsymbol{\epsilon}_\theta(\mathbf{x}_T^d, T)\right\|,
\end{aligned} \tag{27}$$

where the second equivalence holds due to $t = T$ and $\mathbf{x}_T^d = \mathbf{x}_T^a - k_T\boldsymbol{\epsilon}_a = \mathbf{x}_T^a$. Considering that

$$\begin{aligned}
L_b &= \mathbb{E}_{\boldsymbol{\epsilon},t}\left[\left\|\frac{\sqrt{1-\bar{\alpha}_t}}{\sqrt{\bar{\alpha}_t}} \cdot \frac{\bar{\alpha}_t}{1-\bar{\alpha}_t}\boldsymbol{\epsilon}_a + \boldsymbol{\epsilon} - \boldsymbol{\epsilon}_\theta(\mathbf{x}_t^d, t)\right\|^2\right] \\
&= \frac{1}{T}\sum_{t=0}^{T}\mathbb{E}_{\boldsymbol{\epsilon}}\left[\left\|\frac{\sqrt{1-\bar{\alpha}_t}}{\sqrt{\bar{\alpha}_t}} \cdot \frac{\bar{\alpha}_t}{1-\bar{\alpha}_t}\boldsymbol{\epsilon}_a + \boldsymbol{\epsilon} - \boldsymbol{\epsilon}_\theta(\mathbf{x}_t^d, t)\right\|^2\right] \\
&\leq \delta,
\end{aligned} \tag{28}$$

thus

$$\mathbb{E}_{\boldsymbol{\epsilon}}\left[\left\|\frac{\sqrt{\bar{\alpha}_T}}{\sqrt{1-\bar{\alpha}_T}}\boldsymbol{\epsilon}_a + \boldsymbol{\epsilon} - \boldsymbol{\epsilon}_\theta(\mathbf{x}_T^d, T)\right\|^2\right] \leq T \cdot \delta \tag{29}$$

Then we have

$$\begin{aligned}
\mathbb{E}_{\boldsymbol{\epsilon}}\left[\|\hat{\mathbf{x}}_0 - \mathbf{x}_0\|^2\right] &= \mathbb{E}_{\boldsymbol{\epsilon}}\left[\frac{1-\bar{\alpha}_T}{\bar{\alpha}_T}\left\|\frac{\sqrt{\bar{\alpha}_T}}{\sqrt{1-\bar{\alpha}_T}}\boldsymbol{\epsilon}_a + \boldsymbol{\epsilon} - \boldsymbol{\epsilon}_\theta(\mathbf{x}_T^d, T)\right\|^2\right] \\
&\leq \frac{(1-\bar{\alpha}_T)T}{\bar{\alpha}_T}\delta.
\end{aligned} \tag{30}$$

$\square$

### C.3 THE PROOF OF THEOREM 2

**Theorem 2.** *Denote the probability of reversing the adversarial example to the clean example using ADBM and DiffPure as $P(B)$ and $P(D)$, respectively. Then $P(\cdot)$ can be computed as $P(\cdot) = \int \mathbb{1}_{\{\mathbf{x}_0 \notin \mathbb{D}_a\}} p(\mathbf{x}_0 | \hat{\mathbf{x}}_t) \mathrm{d}\mathbf{x}_0$, where $\mathbb{D}_a$ denotes the set of adversarial examples. If the timestep is infinite, the following inequality holds:*

$$P(B) > P(D),$$

*wherein*

$$\text{for } P(B): p(\mathbf{x}_0 | \hat{\mathbf{x}}_t) \propto \exp\left(-\frac{\|\mathbf{x}_t^d - \sqrt{\bar{\alpha}_t}\mathbf{x}_0^a\|^2}{2(1-\bar{\alpha}_t)}\right), \tag{31}$$

$$\text{for } P(D): p(\mathbf{x}_0 | \hat{\mathbf{x}}_t) \propto \exp\left(-\frac{\|\mathbf{x}_t^a - \sqrt{\bar{\alpha}_t}\mathbf{x}_0\|^2}{2(1-\bar{\alpha}_t)}\right). \tag{32}$$

*Proof.* The concept of infinite timestep can be viewed as dividing a finite length of time into infinitesimal intervals, which corresponds to the situation of the following SDEs proposed by Song et al. (2021). Denoting $\mathbf{w}$ as the standard Wiener process, $\bar{\mathbf{w}}$ as the reverse-time standard Wiener process, and $p_t(\mathbf{x})$ the probability density of $\mathbf{x}_t$, the forward process can be described by

$$\mathrm{d}\mathbf{x} = f(\mathbf{x}, t)\mathrm{d}t + g(t)\mathrm{d}\mathbf{w}, \tag{33}$$

where $f(\mathbf{x}, t)$ and $g(t)$ denote the drift and diffusion coefficients, respectively. Under mild assumptions, the reverse process can be derived from:

$$\mathrm{d}\mathbf{x} = [f(\mathbf{x}, t)\mathrm{d}t - g(t)^2 \nabla_{\mathbf{x}} \log p_t(\mathbf{x})]\mathrm{d}t + g(t)\mathrm{d}\bar{\mathbf{w}}. \tag{34}$$

In this context, the reverse of a diffusion process is also a diffusion process, running backwards in time and given by the reverse-time SDE (Eq. (34)). Therefore, if timestep is infinite, $\{\mathbf{x}_t\}_{t:0\to T}$ and $\{\hat{\mathbf{x}}_t\}_{t:T\to 0}$, as the solutions of Eq. (33) and Eq. (34) respectively, follow the same distribution. And due to the Bayes' rule,

$$\begin{aligned} p(\mathbf{x}_0 | \hat{\mathbf{x}}_t) &= p(\mathbf{x}_0) \frac{p(\hat{\mathbf{x}}_t | \mathbf{x}_0)}{p(\hat{\mathbf{x}}_t)} \\ &= p(\mathbf{x}_0) \frac{p(\mathbf{x}_t | \mathbf{x}_0)}{p(\hat{\mathbf{x}}_t)} \\ &\propto p(\mathbf{x}_0) p(\mathbf{x}_t | \mathbf{x}_0) \end{aligned} \tag{35}$$

Since $\hat{\mathbf{x}}_t := \mathbf{x}_t^d$ in ADBM and $\hat{\mathbf{x}}_t := \mathbf{x}_t^a$ in DiffPure, then for all the examples,

$$\begin{aligned} P(B) &= \int \mathbb{1}_{\{\mathbf{x}_0 \notin \mathbb{D}_a\}} p(\mathbf{x}_0 | \hat{\mathbf{x}}_t) \mathrm{d}\mathbf{x}_0 \\ &\propto \int \mathbb{1}_{\{\mathbf{x}_0 \notin \mathbb{D}_a\}} p(\mathbf{x}_0) p(\mathbf{x}_t^d | \mathbf{x}_0) \mathrm{d}\mathbf{x}_0 \\ &\propto \int \mathbb{1}_{\{\mathbf{x}_0 \notin \mathbb{D}_a\}} \exp\left(-\frac{\|\mathbf{x}_t^a - \sqrt{\bar{\alpha}_t}\mathbf{x}_0 - (\sqrt{\bar{\alpha}_t} - k_t)\boldsymbol{\epsilon}_a\|^2}{2(1-\bar{\alpha}_t)}\right) p(\mathbf{x}_0) \mathrm{d}\mathbf{x}_0 \end{aligned} \tag{36}$$

and

$$\begin{aligned} P(D) &= \int \mathbb{1}_{\{\mathbf{x}_0 \notin \mathbb{D}_a\}} p(\mathbf{x}_0 | \hat{\mathbf{x}}_t) \mathrm{d}\mathbf{x}_0 \\ &\propto \int \mathbb{1}_{\{\mathbf{x}_0 \notin \mathbb{D}_a\}} p(\mathbf{x}_0) p(\mathbf{x}_t^a | \mathbf{x}_0) \mathrm{d}\mathbf{x}_0 \\ &\propto \int \mathbb{1}_{\{\mathbf{x}_0 \notin \mathbb{D}_a\}} \exp\left(-\frac{\|\mathbf{x}_t^a - \sqrt{\bar{\alpha}_t}\mathbf{x}_0\|^2}{2(1-\bar{\alpha}_t)}\right) p(\mathbf{x}_0) \mathrm{d}\mathbf{x}_0, \end{aligned} \tag{37}$$

where the $\mathbb{D}_a$ represents the set of adversarial examples. Subtract Eq. (36) by Eq. (37), and we have

$$\begin{aligned} P(B) - P(D) \propto \int \mathbb{1}_{\{\mathbf{x}_0 \notin \mathbb{D}_a\}} p(\mathbf{x}_0) \\ \left[\exp\left(-\frac{\|\mathbf{x}_t^a - \sqrt{\bar{\alpha}_t}\mathbf{x}_0 - (\sqrt{\bar{\alpha}_t} - k_t)\boldsymbol{\epsilon}_a\|_2^2}{2(1-\bar{\alpha}_t)}\right) - \exp\left(-\frac{\|\mathbf{x}_t^a - \sqrt{\bar{\alpha}_t}\mathbf{x}_0\|_2^2}{2(1-\bar{\alpha}_t)}\right)\right] \mathrm{d}\mathbf{x}_0. \end{aligned} \tag{38}$$

Therefore, we only need to compare $\|\mathbf{x}_t^a - \sqrt{\bar{\alpha}_t}\mathbf{x}_0 - (\sqrt{\bar{\alpha}_t} - k_t)\boldsymbol{\epsilon}_a\|_2^2$ and $\|\mathbf{x}_t^a - \sqrt{\bar{\alpha}_t}\mathbf{x}_0\|_2^2$ for comparing $P(B)$ and $P(D)$.

Since $\mathbf{x}_t^a = \sqrt{\bar{\alpha}_t}\mathbf{x}_0^a + \sqrt{1 - \bar{\alpha}_t}\boldsymbol{\epsilon}$ and $\mathbf{x}_0^a = \mathbf{x}_0 + \boldsymbol{\epsilon}_a$,

$$\|\mathbf{x}_t^a - \sqrt{\bar{\alpha}_t}\mathbf{x}_0 - (\sqrt{\bar{\alpha}_t} - k_t)\boldsymbol{\epsilon}_a\| = \|\sqrt{\bar{\alpha}_t}\mathbf{x}_0^a + \sqrt{1 - \bar{\alpha}_t}\boldsymbol{\epsilon} - \sqrt{\bar{\alpha}_t}\mathbf{x}_0 - (\sqrt{\bar{\alpha}_t} - k_t)\boldsymbol{\epsilon}_a\|$$
$$= \|\sqrt{1 - \bar{\alpha}_t}\boldsymbol{\epsilon} + k_t\boldsymbol{\epsilon}_a\|, \tag{39}$$

$$\|\mathbf{x}_t^a - \sqrt{\bar{\alpha}_t}\mathbf{x}_0\| = \|\sqrt{\bar{\alpha}_t}\mathbf{x}_0^a + \sqrt{1 - \bar{\alpha}_t}\boldsymbol{\epsilon} - \sqrt{\bar{\alpha}_t}\mathbf{x}_0\|$$
$$= \|\sqrt{1 - \bar{\alpha}_t}\boldsymbol{\epsilon} + \sqrt{\bar{\alpha}_t}\boldsymbol{\epsilon}_a\|. \tag{40}$$

Since $0 < k_t < \sqrt{\bar{\alpha}_t}$ always holds, thus in expectation we have:

$$\mathbb{E}_{\boldsymbol{\epsilon}}\|\sqrt{1 - \bar{\alpha}_t}\boldsymbol{\epsilon} + k_t\boldsymbol{\epsilon}_a\| < \mathbb{E}_{\boldsymbol{\epsilon}}\|\sqrt{1 - \bar{\alpha}_t}\boldsymbol{\epsilon} + \sqrt{\bar{\alpha}_t}\boldsymbol{\epsilon}_a\|. \tag{41}$$

Hence,

$$\mathbb{E}_{\boldsymbol{\epsilon}}\left[\exp\left(-\frac{\|\mathbf{x}_t^a - \sqrt{\bar{\alpha}_t}\mathbf{x}_0 - (\sqrt{\bar{\alpha}_t} - k_t)\boldsymbol{\epsilon}_a\|_2^2}{2(1 - \bar{\alpha}_t)}\right)\right] > \mathbb{E}_{\boldsymbol{\epsilon}}\left[\exp\left(-\frac{\|\mathbf{x}_t^a - \sqrt{\bar{\alpha}_t}\mathbf{x}_0\|_2^2}{2(1 - \bar{\alpha}_t)}\right)\right], \tag{42}$$

implying that the probability of ADBM reversing the adversarial example to the clean example is higher than that of DiffPure. $\qquad\square$

## D  ADDITIONAL SETTINGS

### D.1  ADDITIONAL TRAINING SETTINGS OF ADBM

We implemented ADBM based loosely on the original implementations by Song et al. (2021). Note that when fine-tuning the ADBM with the pre-trained checkpoint, following Song et al. (2021), we also adopted the continuous version of Eq. (9). This version is conceptually similar to its discrete counterpart, with the exception that $t$ in Eq. (9) represents a continuous value rather than a discrete one. Moreover, we used the Adam optimizer (Kingma & Ba, 2015) and incorporated the exponential moving average of models, with the average rate being 0.999. The batch size was set to 128 for SVHN and CIFAR-10, 112 for Tiny-ImageNet, and 64 for ImageNet-100 (due to memory constraints). All experiments were run using PyTorch 1.12.1 and CUDA 11.3 on 4 NVIDIA 3090 GPUs.

### D.2  ADDITIONAL ADAPTIVE ATTACK CONFIGURATIONS

When performing the adaptive attacks (PGD with EOT) for the AP methods, we set the step sizes to be 0.007, 0.5, and 0.005 for $l_\infty$, $l_1$, and $l_2$ attacks, respectively. For the $l_1$ attack, we set the sparsity level as 0.95. As shown in Sec. 3, the adaptive evaluating is quite time-consuming. Thus following Nie et al. (2022), we conducted the adaptive attacks three times on a subset of 512 randomly sampled images from the test dataset and reported the mean accuracy along with the standard deviation. We note that sometimes the standard deviation is omitted to present the results more clearly.

Table A2: Accuracies (%) of DiffPure under two attacks using the full gradient on CIFAR-10.

| Attack Method | Clean Acc | Robust Acc | | | |
|---|---|---|---|---|---|
| | | $l_\infty$ norm | $l_1$ norm | $l_2$ norm | Average |
| PGD + EOT (Full Grad.) | $92.5 \pm 0.5$ | $42.2 \pm 2.1$ | $44.3 \pm 1.3$ | $60.8 \pm 2.3$ | $49.1 \pm 1.7$ |
| AutoAttack + EOT (Full Grad.) | | 62.70 | 53.91 | 63.87 | 60.16 |

## E  ADDITIONAL EVALUATION RESULTS

### E.1  COMPARISON BETWEEN DIFFERENT ATTACK METHODS ON CIFAR-10

Similar to Lee & Kim (2023), we first performed a comparison between PGD with 20 EOT samples (full gradient) and AutoAttack with the same number of EOT samples (the default setting of `rand`

Table A3: Accuracies (%) of DiffPure and ADBM under different attacks in the $l_2$ setting using the full gradient on CIFAR-10. The strongest attack results for each defense method are highlighted.

| Method | C&W + EOT | DeepFool + EOT | AutoAttack + EOT | PGD + EOT (Our Setting) |
|---|---|---|---|---|
| DiffPure | 74.8 | 78.3 | 63.9 | **60.8** |
| ADBM (Ours) | 78.3 | 84.8 | 66.8 | **63.3** |

Table A4: Accuracies (%) of methods under different white-box adaptive attack threats on SVHN. The same conventions are used as in Tab. 3.

| Architecture | Method | Type | Clean Acc | Robust Acc | | | |
|---|---|---|---|---|---|---|---|
| | | | | $l_\infty$ norm | $l_1$ norm | $l_2$ norm | Average |
| WRN-28-10 | Vanilla | - | 98.11 | 0.19 | 0.02 | 0.03 | 0.08 |
| WRN-28-10 | (Rade et al., 2022) | | 94.46 | 52.65 | 0.16 | 6.76 | 19.86 |
| WRN-28-10 | (Yang et al., 2023) | AT | 93.00 | 52.70 | 0.04 | 3.27 | 18.67 |
| WRN-28-10 | (Wang et al., 2023) | | 95.56 | **64.00** | 0.14 | 5.07 | 23.07 |
| UNet+WRN-28-10 | DiffPure (Nie et al., 2022) | AP | $93.9 \pm 0.7$ | $39.7 \pm 2.2$ | $46.1 \pm 2.1$ | $63.3 \pm 0.8$ | $49.7 \pm 1.7$ |
| UNet+WRN-28-10 | ADBM (Ours) | | $93.5 \pm 0.8$ | $47.9 \pm 1.4$ | $\mathbf{51.2 \pm 0.6}$ | $\mathbf{65.7 \pm 1.5}$ | $\mathbf{54.9 \pm 1.1}$ |

Table A5: Accuracies (%) of methods under different adaptive attack threats on Tiny-ImageNet. The same conventions are used as in Tab. 3.

| Architecture | Method | Type | Clean Acc | Robust Acc | | | |
|---|---|---|---|---|---|---|---|
| | | | | $l_\infty$ norm | $l_1$ norm | $l_2$ norm | Average |
| WRN-28-10 | Vanilla | - | 69.49 | 0.06 | 2.09 | 0.08 | 0.74 |
| WRN-28-10 | (Rade et al., 2022) | | 50.94 | 29.13 | 20.15 | 24.54 | 24.61 |
| WRN-28-10 | (Yang et al., 2023) | AT | 50.89 | 27.02 | 17.79 | 23.29 | 22.70 |
| WRN-28-10 | (Wang et al., 2023) | | 57.59 | **38.41** | 10.73 | 27.21 | 25.45 |
| UNet+WRN-28-10 | DiffPure (Nie et al., 2022) | AP | $58.0 \pm 1.7$ | $24.8 \pm 1.8$ | $44.3 \pm 0.3$ | $32.9 \pm 1.1$ | $34.0 \pm 0.8$ |
| UNet+WRN-28-10 | ADBM (Ours) | | $59.6 \pm 1.2$ | $29.3 \pm 1.7$ | $\mathbf{46.0 \pm 0.4}$ | $\mathbf{38.1 \pm 1.3}$ | $\mathbf{37.8 \pm 0.9}$ |

version AutoAttack yet with full gradient). The main difference between AutoAttack and vanilla PGD lies in the automatic adjustment of step size by AutoAttack, but the adjustment algorithm may be influenced by the randomness of stochastic pre-processing defense. The comparison results are presented in Tab. A2. PGD with EOT achieved significantly better attack performance across all threat models.

We additionally performed extra attack evaluation with the full-gradient C&W (Carlini & Wagner, 2017) and DeepFool (Moosavi-Dezfooli et al., 2016) attacks, which are optimization-based attacks that differ significantly from PGD. Here their original $l_2$ setting (Carlini & Wagner, 2017; Moosavi-Dezfooli et al., 2016) were evaluated with 1,000 iteration steps and 20 EOT samples. The implementation used the standard `torchattacks` benchmark[4]. The robustness results are shown in Tab. A3 (along with the PGD and AutoAttack $l_2$ results in Tab. A2). We can see that PGD with EOT is also significantly stronger than other attacks such as C&W and DeepFool, which need considerably more iteration steps. Additionally, ADBM is more robust than DiffPure regardless of the evaluation methods.

Thus, considering that AutoAttack is widely recognized as a reliable attack method for AT models, we employed AutoAttack for AT evaluation. Conversely, for AP evaluation, we measured the worst-case robustness of each defense method by employing PGD with EOT in our main experiments.

E.2 ADDITIONAL WHITE-BOX RESULTS ON SEVERAL DATASETS

The accuracies on SVHN are shown in Tab. A4. ADBM consistently achieved significantly better adversarial robustness than DiffPure across all adversarial threats. On average, ADBM outperformed DiffPure by 5.2%. In addition, although SOTA AT models showed excellent performance on seen

---

[4]https://github.com/Harry24k/adversarial-attacks-pytorch

Table A6: Accuracies (%) of methods under different adaptive attack threats on ImageNet-100. The same conventions are used as in Tab. 3.

| Architecture | Method | Type | Clean Acc | Robust Acc | | | |
|---|---|---|---|---|---|---|---|
| | | | | $l_\infty$ norm | $l_1$ norm | $l_2$ norm | Average |
| ResNet-50 | (Liu et al., 2023a) | AT | 78.8 | 47.2 | 3.6 | 25.0 | 25.3 |
| ResNet-50 | (Debenedetti et al., 2023) | | 79.5 | **50.3** | 3.9 | 23.8 | 26.0 |
| UNet+ResNet-50 | DiffPure (Nie et al., 2022) | AP | 77.3 | 22.2 | 55.5 | 58.4 | 45.4 |
| UNet+ResNet-50 | ADBM (Ours) | | 79.5 | 25.2 | **58.6** | **61.3** | **48.4** |

Table A7: Accuracies (%) of methods under three query-based attacks and the transfer-based attack on CIFAR-10. The same conventions are used as in Tab. 4.

| Architecture | Method | Type | Clean Acc | Robust Acc | | | | |
|---|---|---|---|---|---|---|---|---|
| | | | | RayS | Square | SPSA | Transfer | Average |
| WRN-70-16 | Vanilla | - | 97.02 | 6.64 | 2.44 | 1.37 | - | - |
| WRN-70-16 | (Gowal et al., 2020) | | 91.10 | 79.20 | 71.62 | 80.76 | 89.98 | 82.33 |
| WRN-70-16 | (Rebuffi et al., 2021) | AT | 88.54 | 73.91 | 69.35 | 75.82 | 87.64 | 79.05 |
| WRN-70-16 | (Gowal et al., 2021) | | 88.74 | 78.81 | 73.93 | 79.88 | 87.89 | 81.85 |
| WRN-70-16 | (Wang et al., 2023) | | 93.25 | 81.89 | 76.30 | 82.13 | 90.34 | 84.78 |
| UNet+WRN-70-16 | (Yoon et al., 2021) | | 87.93 | 74.22 | 72.17 | 68.55 | 84.28 | 77.43 |
| UNet+WRN-70-16 | DiffPure (Nie et al., 2022) | AP | 92.51 | 90.92 | 90.53 | 90.33 | 90.16 | 90.89 |
| UNet+WRN-70-16 | ADBM (Ours) | | 91.86 | **91.31** | **91.50** | **90.72** | **90.39** | **91.16** |

attacks, they struggle to exhibit robustness against unseen $l_1$ and $l_2$ attacks. On average, ADBM surpasses these AT models by 31.8%.

The white-box adaptive evaluation results on Tiny-ImageNet are respectively presented in Tab. A5 and Tab. A6. In the Tiny-ImageNet evaluation, we maintained consistent settings with those used for CIFAR-10, with the exception that for the $l_\infty$ threat, we set the bound to be $\epsilon_\infty = 4/255$, which is a popular setting for this dataset (Yang et al., 2023; Wang et al., 2023). In the ImageNet-100 evaluation, we set the bounds $\epsilon_\infty = 4/255$, $\epsilon_1 = 75$, $\epsilon_2 = 2$, which is a commonly used setting for the high-resolution datasets like ImageNet-1K. For this dataset, we directly compared ADBM with representative ResNet-50 AT checkpoints specified trained on ImageNet-1K by masking out the 900 output channels not included in ImageNet-100. By examining Tab. A5 and Tab. A6, we can draw similar conclusions to those observed on SVHN and CIFAR-10.

### E.3 ADDITIONAL RESULTS UNDER BLACK-BOX SEEN THREATS

The evaluation results on CIFAR-10 under black-box seen threats are presented in Tab. A7. In this evaluation, we maintained consistent settings with those used for SVHN. By examining Tab. 4 and Tab. A7, we can conclude that under the realistic black-box attacks, ADBM has advantages over AT models even on the seen threat.

### E.4 RESULTS UNDER BLACK-BOX UNSEEN THREATS

We further evaluate the generalization ability of ADBM on more unseen threats beyond norm-bounded attacks. Two patch-like attacks, PI-FGSM (Gao et al., 2020) and NCF (Yuan et al., 2022), and two recent diffusion-based attacks, DiffAttack (Chen et al., 2023b) and DiffPGD (Xue et al., 2023) were evaluated. As the code of these methods only natively supports input resolution $224 \times 224$, we conducted experiments on ImageNet-100. The results are show in Tab. A8. We can conclude that ADBM demonstrates better generalization ability than DiffPure and previous AT methods against these black-box unseen threats.

Table A8: Accuracies (%) of methods under different black-box unseen threats on ImageNet-100.

| Architecture | Method | Type | NCF | PI-FGSM | Diff-Attack | Diff-PGD |
|---|---|---|---|---|---|---|
| ResNet-50 | (Liu et al., 2023a) | AT | 3.8 | 26.3 | 21.5 | 29.4 |
| ResNet-50 | (Debenedetti et al., 2023) | | 11.2 | 29.4 | 28.5 | 49.8 |
| UNet+ResNet-50 | DiffPure | AP | 30.1 | 38.4 | 39.8 | 71.9 |
| UNet+ResNet-50 | ADBM (Ours) | | **33.9** | **42.7** | **42.8** | **72.4** |

Table A9: Accuracies (%) of different methods on CIFAR-10. Here the classifier used a WRN-28-10 or a vision transformer. ADBM (Transfer) was trained with noise from a WRN-70-16 while ADBM (Direct) was trained with noise from the WRN-28-10 or the vision transformer directly.

| Method | Classifier | $l_\infty$ | $l_1$ | $l_2$ | Average |
|---|---|---|---|---|---|
| DiffPure | | 42.4 | 60.3 | 43.8 | 48.8 |
| ADBM (Transfer) | WRN-28-10 | 45.1 | 61.7 | **52.0** | 52.9 |
| ADBM (Direct) | | **45.3** | **62.1** | 51.9 | **53.1** |
| DiffPure | | 25.0 | 41.8 | 51.6 | 39.5 |
| ADBM (Transfer) | ViT | 28.9 | 46.7 | 54.1 | 43.2 |
| ADBM (Direct) | | **30.9** | **48.4** | **56.1** | **45.1** |

Table A10: Inference time (in seconds) for diffusion-based purification on different datasets, corresponding to the reverse sampling (RS) steps using an NVIDIA 3090 GPU (with a batch size of 1).

| Dataset | Network | RS = 1 | 2 | 3 | 4 | 5 | 100 (Original DiffPure) |
|---|---|---|---|---|---|---|---|
| CIFAR-10 | UNet+WRN-70-16 | 0.060 | 0.099 | 0.152 | 0.189 | 0.248 | 4.420 |
| SVHN | UNet+WRN-28-10 | 0.052 | 0.095 | 0.139 | 0.182 | 0.229 | 4.349 |
| Tiny-ImageNet | UNet+WRN-28-10 | 0.056 | 0.010 | 0.149 | 0.195 | 0.232 | 4.407 |

### E.5 RESULTS OF ADBM ON NEW CLASSIFIERS

Here we utilized the fine-tuned ADBM checkpoint, trained with adversarial noise from a WRN-70-16 classifier, as the pre-processor for a WRN-28-10 model and a vision transformer (ViT) model[5] directly, denoted as ADBM (Transfer). The results shown in Tab. A9 demonstrate that ADBM (Transfer) achieved robustness levels comparable to ADBM trained directly with WRN-28-10 or the ViT (52.9% vs. 53.1% on WRN-28-10, and 43.2% vs. 45.1% on ViT). This finding highlights the practicality of ADBM, as the fine-tuned ADBM model on a specific classifier can potentially be directly applied to a new classifier without requiring retraining. We guess this may be attributed to the transferability of adversarial noise (Dong et al., 2018).

## F INFERENCE COST OF ADBM

On one hand, the inference cost of diffusion-based purification can be significantly reduced via fast sampling according to our investigation. Tab. A10 gives the actual inference time (in seconds) for diffusion-based purification on different datasets, corresponding to the reverse sampling steps. Given that ADBM has demonstrated effectiveness even with a single reverse step (Tab. 2), it has potential applications in various real-time scenarios (only 0.06s is needed for inference in this case, as shown in Tab. A10).

On the other hand, similar to DiffPure, ADBM necessitates additional parameters. Taking CIFAR-10 as an example, the classifier WRN-70-16 consists of 267M parameters, while the UNet architecture, specifically the `DDPM++ continuous` used in this work, consists of 104M parameters. It remains an open question whether the scale of the UNet architecture, which is currently directly adapted from architectures designed for generative tasks, can be further reduced for adversarial purification. We leave it to future work.

---

[5]We used the checkpoint released by `https://github.com/dqj5182/vit_cnn_cifar_10_from_scratch`, which had 90.6% clean accuracy on CIFAR-10.

## G  BOARDER IMPACT

Our method, which effectively purifies adversarial noise, can potentially hinder the utility of certain "adversarial for good" methods. For example, it could cause more difficult copyright protection using adversarial noise (Liang et al., 2023) when abusing our method. In addition, there seems to be a trade-off between adversarial robustness and the risk of privacy leakage (Li et al., 2024a). But in general, we believe the concrete positive impacts on trustworthy machine learning outweigh the potential negative impacts.

