# OpenReview forum: "ADBM: Adversarial Diffusion Bridge Model for Reliable Adversarial Purification"
_ICLR.cc/2025/Conference — ICLR 2025 Poster_

### Official Review · Reviewer_pfC6 · 2024-10-28

**Soundness:** 3
**Presentation:** 2
**Contribution:** 3
**Rating:** 8
**Confidence:** 4

**Summary:**

Adversarial attack is a challenge for the neural network. This paper proposed a diffusion model-based adversarial purification (AP) to defend against adversarial attacks via adversarial training (AT) called ADBM. ADBM alleviates the trade-off between noise purification efficiency and recovery quality by proposing a new loss function to fine-tune the diffusion model via the AT paradigm. ADBM also offers detailed proofs to ensure the adversarial perturbation will be gradually removed. The experimental results show that ADBM achieves the SOTA performance.

**Strengths:**

1. This paper proposed a new adaptive pipeline based on the diffusion model's full gradient, which can further ensure the diffusion-based AP's robustness. The robustness of the diffusion-based AP is always questionable since it is a computation cost to get the full gradient of the diffusion model. Previous works proposed various gradient estimation methods, such as the adjoint method from DiffPure and the surrogate process from Lee. et al. [1], which remains a question: Is the improvement of such diffusion-based AP due to the estimation error of the gradient? Lee. et al. also prove this question. This paper is the first work to evaluate the robustness of diffusion-based AP by the full gradient to the best of my knowledge. I think this will benefit further study of diffusion-based AP.

2. The AT paradigm for diffusion-based AP proposed in this paper is novel and robust. To establish the loss function for AT, this paper chooses to connect the adversarial samples and the generation target $x_{0}$, which is interesting. Then, the detailed proofs verify that the loss function could ensure the removal of the adversarial perturbation after various times of the reverse process.

3. The experimental results show that ADBM achieves the SOTA performance. Specifically, the experiment also includes unseen attacks to illustrate how the proposed AT paradigm could work on unseen threats.

[1] Minjong Lee and Dongwoo Kim. Robust evaluation of diffusion-based adversarial purification. In Int. Conf.
Comput. Vis. (ICCV), 2023

**Weaknesses:**

1. The writing for this paper should be improved. 1) Some math symbols are confused. For example $\hat{x}\_0$. It first indicated in 299 lines that $\hat{x}\_0$ should be the final point of the reverse process, thus having $\hat{x}\_0=x\_{0}$. But, in 879 lines, $\hat{x}\_0$ changed to the prediction of $x_{0}$ for $x_{t}$, thus causing the confusion. There are various similar mistakes, such as $x_{t}^{d}$ in 265 lines.  2) One-step DDIM is confusing since this paper includes the discrete and continuous VP-SDE. Setting $s=1$ is truly the one-step DDIM based on the discrete VP-SDE but is not one-step for continuous VP-SDE since the time sequence of continuous VP-SDE is in $[0.0, 1.0]$. In this condition, $s=1$ means the entire reverse process for continuous VP-SDE instead of the one-step. The authors should explicitly clarify which formulation (discrete or continuous) they use when discussing one-step DDIM and explain the implications for both cases.

2. The experimental results may not be enough. 1) The BPDA [1] should be considered to test the performance of the ADBM since it is a special attack designed for the AP.  Adding additional experiments on CIFAR-10 against BPDA ($\ell_{\infty}$,$\epsilon = 8/255$) will further strength the contribution. 2) AToP should be considered as the baseline, even if only making an ablation study on CIFAR-10 since ADBM is similar to AToP. The only difference is that AToP may be suitable for any generative model, and ADBM is for the diffusion model only.  Adding additional experiments on CIFAR-10 against PGD+EOT ($\ell_{\infty}$,$\epsilon = 8/255$, EOT=20, steps=200) to compare the performance of ADBM and AToP (the generative model chooses GAN, which is the best reported in AToP) will enhance the contribution for this paper. Additionally, adding an ablation study between ADBM and AToP (the generative model chooses VP-SDE) against PGD+EOT ($\ell_{\infty}$,$\epsilon = 8/255$, EOT=20, steps=200) will further show the validity of ADBM.

[1] Minjong Lee and Dongwoo Kim. Robust evaluation of diffusion-based adversarial purification. In Int. Conf.
Comput. Vis. (ICCV), 2023

**Questions:**

1. Does ADBM only work for the DDIM sampler? The motivation for this question is that the assumption for the proofs is the DDIM sampler. What happens if we change the DDIM sampler to the original DDPM sampler?

To sum up, there are two main contributions for ADBM: 1) proposed a full-gradient adaptive attack method. 2) proposed an interesting AT paradigm to enhance AP based on the unconditional diffusion model. All my concerns are listed in Weaknesses and Questions. Considering 1) the lacking the necessary baseline and attack method and 2) the writing problem, I rate it as "marginally above the acceptance threshold." If the author could solve these concerns, I'm willing to increase my rate.

---

> ### Author Response · Authors · 2024-11-22
> **Thank you for the valuable review (1/2)**
>
> Thank you for the supportive review. We are encouraged by the appreciation of the comprehensive evaluation, the novelty and the detailed proof of this work. We have uploaded a revision of our paper to improve the writing (**revised part is shown in blue**, additional results will be added soon). Below we address the detailed comments, and hope that you may find our response satisfactory.
>
> **Q1. About the writing for this paper.**
>
> Thank you again for your careful review, which helps us imporve the quality of this paper. We have checked our formulations detailedly and made a revision of our draft (see the updated pdf). In addition, we have the following clarifications:
>
> **Q1.1 Some math symbols are confused, for example $\hat{x}_0$. In line 229 $\hat{x}_0$ is used to denote the final point of the reverse process, thus having $\hat{x}_0 = x_0$, but in line 879 $\hat{x}_0$  is changed to the prediction of $x_0$ for ${x}_t$, thus causing the confusion.**
>
> We apologize for the confusion.  We believe that the confusion may come from the facts that 1) the output of reverse process should be the clean sample, i.e., $\hat{x}_0:={x}_0$, and 2) only one reverse step is used and thus the reversed output $\hat{x}_0$ is indeed the prediction of $x_0$ from $x_t^a$.
>
> More specifically, $\hat{x}_{t:T \to 0}$ is denoted as the different states in the reverse Markov chain (line 226). In line 229, $\hat{x}_0$, the final point of the reverse process (the reverse Markov chain), is expected to directly predict $\hat{x}_0$, i.e. $\hat{x}_0:={x}_0$, so as to achieve the goal of purification. And in line 879, since we employ a one-step reverse process, it is appropriate to reuse the symbol of $\hat{x}_0$, the final point of the reverse process, to denote the prediction of $x_0$ from $x_t^a$.
>
> The situation is as same as $x_t^d$. It is the final point of the forward process of ADBM, as both adversarial noise and Guassian noise are added. On the other hand, we purify them simultaneously in the reverse process, which begin with $x_t^d$. Given $\hat{x}_t := x_t^d = x_t^a-k_t\epsilon_a, k_0=1, k_T=0$ (line 231), we have $x_t^d=x_t^a$ when $t=T$, which is the end of the forward process and the start of the reverse process.
>
> We have made a revision to make it clearer.
>
> **Q1.2 One-step DDIM is confusing since this paper includes the discrete and continuous VP-SDE.**
>
> Thanks for pointing out this. In the discrete case, ${\tau_0,\ldots,\tau_s}$ is a linearly increasing sub-sequence of $\{0,\ldots,T\}$, $\tau_0=1,\tau_s=T$. In the continuous case, $\{\tau_0,\ldots,\tau_s\}$ is a linearly increasing sequence of $[0,T] \sub [0.0,1.0]$, $\tau_0=1,0\leq\tau_s=T\leq1$.  $T$ is determined in the forward process. We have made a revision to discuss the two cases respectively.
>
> **Q2. About experimental results**
>
> **Q2.1. About the BPDA results.**
>
> Thanks for your valuable suggestion. We additionally evaluated DiffPure and ADBM on CIFAR-10 in the BPDA attack setting, and the results are shown below (200 iterations with 20 EoT steps):
>
> |||||
> |-|-|-|-|
> ||Linf|L1|L2|
> |DiffPure|69.53|71.29|80.08|
> |ADBM|70.51|72.07|80.47|
>
>
> We can see that 1) ADBM obtains improved robustness over DiffPure; 2) BPDA is a weak attack for diffusion-based purification compared with the evaluation using full gradient (Table 3). The second point has also been validated in RQ3 of [1]. We will add these results to the revised version.
>
> [1] Minjong Lee and Dongwoo Kim. Robust evaluation of diffusion-based adversarial purification. ICCV, 2023

---

> ### Author Response · Authors · 2024-11-22
> **Thank you for the valuable review (2/2)**
>
> **Q2.2. Comparison with AToP**
>
> Since AToP is not a diffusion-based purification method, we did not include it in our initial submission. According to your suggestion, we have contacted the authors of AToP for the trained checkpoint of their work. Using their released code, we successfully replicated the results presented in their paper.
>
> However, by a thorough review of their code implementation, we discovered that the used attacks in the original implmentation of AToP are, in fact, weak attacks. Their evaluation does not employ the full gradient of the entire adversarial perturbation process; instead, it relies solely on BPDA or generates adversarial samples on the original classifier. We further evaluated AToP's robustness under our reliable evaluatiion (linf, 8/255 on CIFAR-10, full gradient, EOT=20, steps=200) and regrettably found that their method loses robustness. **Even $RT_2$ with GAN (the best implementation according to Table 8 of the AToP paper) achieved only 1.3% robust accuracy.**
>
> **Regarding AToP's use of diffusion models, our preliminary findings indicate a 6.7% drop in robustness compared to DiffPure.** This might be attributed to AToP's requirement for a diffusion model capable of handling masked images. But overall, this aligns with previous attempts to improve diffusion-based purification methods. As shown in Table 3, many works [2,3,4] tried to improve diffusion-based purification but also exhibited negative performance compared to the original DiffPure under our reliable evaluation. In contrast, ADBM is, to our best knowledge, the only method that provides actual improvements compared with DiffPure.
>
>
>
> **Q3. The assumption for the proofs is the DDIM sampler. What happens if we change the DDIM sampler to the original DDPM sampler?**
>
> Thanks for pointing out this. We would like to clarify that besides Theorem 1, Theorem 2 indicates that with infinite reverse timesteps, adversarial examples purified with ADBM are more likely to align with the clean data distribution than those with DiffPure, which is independent of the DDIM sampler.
>
> Using the original DDPM sampler will incur significant inference costs proportional to the reverse steps. Our intention of mainly reporting the results using DDIM sampler is to investigate whether robustness can be maintained with minimal cost (i.e., one-step reverse) for both DiffPure and ADBM, which cannot be achieved by DDPM (see Table 2).
>
> But as per your suggestion, we have also evaluated ADBM under the 100 reverse steps using the original DDPM sampler and the results are shown below (CIFAR-10, $l_\infty$ threat, EOT=20, steps=200):
>
> |||
> |-|-|
> ||DDPM (100 reverse steps)|
> |DiffPure|45.8|
> |ADBM|**49.8**|
>
>
> We can see that ADBM also has better robustness over DiffPure. However, it is important to consider the practical implications of this approach. The increase in inference cost associated with 100 reverse steps gives significant practical constraints in real-world applications. In contrast, one of our contributions is demonstrating we can use the DDIM sampler to achieve similar robustness while significant reducing the inference cost by employing the DDIM sampler (line 101).
>
> **We hope you are satisfied with our response above. If you have any further comments, and we'd be glad to answer them in the discussion period.**
>
>
> [2] Jinyi Wang, et al. Guided diffusion model for adversarial purification. arXiv:2205.14969, 2022.
>
> [3] Boya Zhang, et al. Purify++: Improving diffusion-purification with advanced diffusion models and control of randomness. arXiv:2310.18762, 2023
>
> [4] Boya Zhang, et al. Enhancing adversarial robustness via score-based optimization. NeurIPS, 2024.

---

> > ### Comment · Reviewer_pfC6 · 2024-11-24
> > **Response for rebuttal**
> >
> > Thanks for the author`s rebuttal. I have carefully checked all the contents. Firstly, please carefully check the version of the manuscript since the current version is "PATCH-BASED COMPOSITE ADVERSARIAL TRAINING AGAINST PHYSICALLY REALIZABLE ATTACKS ON OBJECT DETECTION," which is irrelevant to the AP and ADBM.
> >
> > Then, the author addresses my concerns about the experiments. An interesting part is that ATOP cannot work for the diffusion models, which further enhances the contribution of this paper. Meanwhile, under the BPDA attack, ADBM is better than Diffpure, which verifies its validity. Due to the problem with the wrong version, I choose to keep my rate temporarily.

---

> > > ### Author Response · Authors · 2024-11-25
> > > **Expecting for further feedback**
> > >
> > > **Dear Reviewer pfC6:**
> > >
> > > Thank you again for your invaluable feedback and for acknowledging our additional experimental results. We would like to inquire whether other questions have been resolved and we look forward to your further comments.

---

> > > > ### Comment · Reviewer_pfC6 · 2024-11-25
> > > > **Response to rebuttal**
> > > >
> > > > Thanks for the further clarification from the author. I have carefully checked all the contents, including the revision of the manuscript.  The new manuscript addresses my concerns about the theory proofs based on the discrete and continuous DDIM samplers and alleviates the confusion about the $\hat{x}\_0$. Although some symbols, such as $x_{T}^{a}$, could be simplified to make the proof more readable, the author addressed most of my concerns.
> > > >
> > > > To sum up, the main contributions of this paper are as follows: 1) Propose the full gradient evaluation for diffusion models based on AP. 2) Propose an AT-based paradigm to enhance diffusion models based-AP, where the author offers detailed proofs to show that AT could finally cancel the adversarial perturbation term hidden on the adversarial samples. 3) The author reports detailed experiments, including unseen attacks, to prove the validity and efficiency of the proposed method. The major weakness of this paper is that the AT paradigm will improve the training computation cost for the diffusion models-based AP, which hinders its implementation on the ImageNet-1k dataset.
> > > >
> > > >  I think the pros outweigh cons, and full gradient evaluation indeed benefits the further study of the diffusion models based-AP. Therefore, I decided to increase the rate to "accept."

---

> > > > > ### Author Response · Authors · 2024-11-28
> > > > >
> > > > > **Dear Reviewer pfC6:**
> > > > >
> > > > > We sincerely appreciate your recognition of our work and your decision to increase the score. We will simplify the notation further in the revised version. Your reviews and feedbacks are quite valuable for helping us strengthen the work further.

---

> ### Author Response · Authors · 2024-11-24
> **Sorry for the revision error (File upload error)**
>
> **Dear Reviewer pfC6:**
>
> We sincerely thank you for your timely response and for recognizing the significant error. This is indeed another irrelevant anonymous file due to our negligence. We have updated the correct version of the manuscript, which includes revisions to improve the writing (the revised sections are highlighted in blue). Please recheck the revised version.
>
> Once again, we appreciate your efforts and recognition of our work.
>
> Sincerely,
> Authors

---

### Official Review · Reviewer_H77B · 2024-10-28

**Soundness:** 2
**Presentation:** 3
**Contribution:** 2
**Rating:** 6
**Confidence:** 5

**Summary:**

This paper proposes a new method for Diffusion-based Purification, which is named as Adversarial Diffusion Bridge Model (ADBM). Different from previous methods, ADBM directly constructs a reverse bridge from the diffused adversarial data back to its original clean examples. Experiments show the effectiveness of the proposed method.

**Strengths:**

The proposed ADBM has the complete theoretical support, and the experiments on the CIFAR10 demonstrate the effectiveness of the proposed method on some chosen datasets.

**Weaknesses:**

1. The proposed method is only verified on the datasets with small classification number, such as CIFAR10 and SVHN. The experiments on datasets with small sizes can not fully demonstrate the effects of the proposed strategy, since the applications in the real world will have large number of classification number. A more suitable dataset could be ImageNet with at least 1K class number. I think the complicated data distribution will largely influence the performance of diffusion models.

2. The experimental results are mainly built with the network of WRN, and the experimental results with more types of networks are not shown.

3. Compared with other methods, the accuracy on the adversarial examples are not improved a lot, while the accuracy on the clean examples are decreased clearly. The authors should analyze the factor which limits the accuracy on the clean examples.

**Questions:**

1. What is the performance on more datasets with more classification numbers?

2. What is the performance of the proposed method with various networks?

3. What is the performance of the proposed method towards different types of attack?

**Details Of Ethics Concerns:**

The proposed method could be utilized to enhance the AI attack methods, which can cause the safety issue.

---

> ### Author Response · Authors · 2024-11-22
> **Thank you for the valuable review (1/2)**
>
> We appreciate your recognition of our theoretical results and the effort devoted to checking this paper. We will add the additional results and clarifications to the revised version. Our responses to the weaknesses and questions are as follows:
>
> *Q1. The proposed method is only verified on datasets with small classification number, such as CIFAR10 and SVHN. The experiments on datasets with small sizes can not fully demonstrate the effects of the proposed strategy, since the applications in the real world will have large number of classification number.  A more suitable dataset could be ImageNet with at least 1K class number.*
>
>
>
> Thanks for your question. On the one hand, we would like to clarify that **beside CIFAR-10 and SVHN, we have evaluated ADBM on more datasets such as ImageNet-100 and Tiny-ImageNet, which have 100 and 200 classes respectively. The results in Table A5 and Table A6 show that ADBM still have good robustness on these cases with more classes (with at least 3% robustness improvements).**
>
> On the other hand, we apologize that a full comparison using ImageNet-1K with 1K classes is beyond our current capabilities. According to the report of  [1], training a diffusion model with good generative ability on the ImageNet with 1K classes required 32xA100 (80G) training for 13 days, even on the resolution of $64 \times 64$. Although our ADBM only needs fine-tuning about 1/10 of the training steps of the original diffusion models, it is still a great burden to us. But as per your suggestion, we have tried our best to perform a preliminary experiments on ImageNet with 1K classes within the rebuttal period. **Here we fine-tuned ADBM with just one epoch on ImageNet and used a ViT-B [2] as a classifier**, and the results are shown in below:
> ||||
> |-|-|-|
> ||Clean|Linf|
> |DiffPure|77.93|38.86|
> |ADBM|**78.52**|**41.02**|
>
>
> These results show that even on complicated data distribution with 1000 classes, ADBM can obtain considerable improments over previous methods. And we believe that the gain will be more significant if more training resources can be provided (note that here we use just one epoch training).
>
> In fact, according to recent advances of text-to-image diffusion models, diffusion models have quite good scalability even on the web-scale image dataset. And thus from this persepctive, the classes and the complicated data distribution could not be a main concern.
>
>
> *Q2. The experimental results are mainly built with the network of WRN, and the experimental results with more types of networks are not shown.*
>
> We use WRN because most of preivous works mainly used this network for evaluation and our intention is to perform a fair comparison. In addition, we would like to clarify that **besides WRN, we have included the results on ViT (vision transformer) in  Table A9 and the results on ResNet-50 in Table A8. On all these networks, together with WRN-70-16 and WRN-28-10**, ADBM show better robustness than previous methods.
>
> [1] [https://github.com/NVlabs/edm](https://github.com/NVlabs/edm)
>
> [2] [https://pytorch.org/vision/stable/models/generated/torchvision.models.vit_b_16.html?highlight=vit+b#torchvision.models.vit_b_16](https://pytorch.org/vision/stable/models/generated/torchvision.models.vit_b_16.html?highlight=vit+b#torchvision.models.vit_b_16)

---

> ### Author Response · Authors · 2024-11-22
> **Thank you for the valuable review (2/2)**
>
> *Q3. Compared with other methods, the accuracy on the adversarial examples are not improved a lot, while the accuracy on the clean examples are decreased clearly. The authors should analyze the factor which limits the accuracy on the clean examples.*
>
> First, **we would like to clarify that compared with DiffPure and other AT methods, ADBM does not decrease the accuracy on clean examples.** On CIFAR-10, the standard deviation of results are about $\pm 0.5$ and the difference between ADBM and DiffPure on clean images are also about $\pm 0.5$ (see Table 3). Besides, on Tiny-ImageNet (see Table A5) and ImageNet-100 (see Table A6), and the additional results on ImageNet (see **our response to Q1**), the accuracy on clean images even slightly increased.
>
> Second, **we respectfully disagree that adversarial robustness did not improve a lot.** On the one hand, **several attempts to improve DiffPure ( see Table 3) have resulted in negative effects compared to the original DiffPure**, which means improve the diffusion-based purification is non-trivial. On the other hand, considering recent efforts to improve adversarial robustness, the 4.4% robustness gain of ADBM on CIFAR-10 is significant. For example, Kuang et al. [3] reported a 1.0% robustness gain under PGD on CIFAR-10 over the baseline method, Zhang et al. [4] achieved a 0.4% robustness gain under PGD on CIFAR-10, and Wu et al. [5] obtained a 1.0% robustness gain on CIFAR-10. These works [3, 4, 5] have all been published in recent ML conferences.
>
>
>
>
>
> *Q4. What is the performance on more datasets with more classification numbers?*
>
> Please see **our response to Q1**.
>
>
>
> *Q5. What is the performance of the proposed method with various networks?*
>
> Please see **our response to Q2**.
>
>
>
> *Q6. What is the performance of the proposed method towards different types of attack?*
>
> Thanks for this question. We have evaluated ADBM on **three attacks with different threats** in Table 3. In additon, we have evaluated ADBM on **four** attacks in Table 4 and Table A7 (configurations detailed in Section 5.3), including four unseen attacks: RayS, SPSA, Transfer, and Square attacks. Furthermore, we have conducted experiments on **four additional unseen** threats including patch-like attacks (PI-FGSM and NCF) and recent diffusion-based attacks (DiffAttack and DiffPGD), and the results are shown in Table A8 (configurations detailed in Appendix E.4).
>
> **Overall, we have evaluated ADBM on at least 11 different types of attacks. We can see that under all these attacks, ADBM demonstrates better performance and generalization ability than DiffPure and previous AT methods. We believe our evaluation is comprehensive.**
>
>
>
>
>
> **If the reviewer agrees with our clarification above and kindly re-evaluates the value of our work, we would be very grateful. We are happy to address any further questions regarding our work.**
>
> [3] Kuang et al. Improving adversarial robustness via information bottleneck distillation. NeurIPS, 2023.
>
> [4] Zhang et al. Improving Accuracy-robustness Trade-off via Pixel Reweighted Adversarial Training, ICML, 2024.
>
> [5] Wu et al. Annealing Self-Distillation Rectification Improves Adversarial Training. ICLR, 2024.

---

> ### Comment · Reviewer_H77B · 2024-11-27
> **Review comments**
>
> After reading the response from the authors, the concerns about the dataset and network diversity have been solved. However, I still think the improvement on the robustness is not obvious. I will change my rating to marginally above the acceptance threshold.

---

> ### Author Response · Authors · 2024-11-28
>
> **Dear Reviewer H77B**:
>
> We sincerely thank you for the timely and positive feedback on our work. It is quite valuable for helping us strengthen this work. We believe that in many security-critical cases, ADBM, with favorable theoretical guarantees, good empirical improvements (**+4.4%** under reliable evaluation), and plug-and-play features, offers an alternative way to further enhance adversarial robustness.

---

### Official Review · Reviewer_hELf · 2024-11-03

**Soundness:** 2
**Presentation:** 3
**Contribution:** 2
**Rating:** 5
**Confidence:** 4

**Summary:**

This paper trains a purifier based on the theoretical framework of a diffusion model, making it better suited for adversarial purification tasks.

**Strengths:**

- Introduces a comprehensive training process to adapt the diffusion model for adversarial purification tasks.
- Theoretically and empirically demonstrates superior performance compared to DiffPure.

**Weaknesses:**

- Unlike DiffPure, the Adversarial Diffusion Model (ADBM) requires reasoning about adversarial perturbations for purification, which may **limit its generalizability to unseen attacks**, similar to challenges faced by adversarial training. For instance, the experimental results in Table 3—beyond different settings of white-box attacks, how does it perform against Square attacks or StAdv?
- The author mentions the Robust Diffusion Classifier (RDC) method in the paper but **does not compare it with the purification component of RDC, namely Likelihood Maximization (LM)**. To my knowledge, RDC’s LM can handle unseen attacks better than ADBM without needing adversarial perturbation training.
- In the original diffusion model framework, as the time step approaches infinity, $x_T$ converges to a standard Gaussian distribution. With ADBM losing this property, there may need to be corresponding adjustments in the selection of variables such as $\bar{\alpha}$.

**Questions:**

please see weaknesses

---

> ### Author Response · Authors · 2024-11-22
> **Thank you for the valuable review**
>
> We appreciate your recognition of our theoretical and empirical results and the effort devoted to checking this paper. We will add the additional results and clarifications to the revised version. Our responses to the weaknesses and questions are as follows:
>
> **_Q1: Generalizability to unseen attacks like Square and StAdv attacks_**
>
> Thanks for this question. Beyond different settings of white-box attacks in Table 3, we have evaluated ADBM on **four unseen** attacks in Table 4 and Table A7 (configurations detailed in Section 5.3), including four unseen attacks: RayS, SPSA, Transfer, and **Square attacks**. Furthermore, we have conducted experiments on **four additional unseen** threats including patch-like attacks (PI-FGSM and NCF) and recent diffusion-based attacks (DiffAttack and DiffPGD), and the results are shown in Table A8 (configurations detailed in Appendix E.4).
>
> Here we additionally provide the results under StAdv according to your suggestion (see Table 4 for results under Square attack):
>
> ||||
> |-|-|-|
> ||StAdv (on SVHN)|StAdv (on CIFAR-10)|
> |DiffPure|89.45|85.94|
> |ADBM|**92.19**|**87.30**|
>
>
> **We can see that under all these unseen attacks, which cover various types adversarial attacks, ADBM demonstrates better performance and generalization ability than DiffPure and previous AT methods.** **We believe our evaluation on unseen attacks is comprehensive.**
>
> **_Q2: About Likelihood Maximization in RDC_**
>
> Thank you for your suggestion. We have discussed the difference between RDC and ADBM in Appendix A.2 while ignoring the Likelihood Maximization (LM). Here we further clarify this:
>
> We agree that LM is indeed a purification defense, which has the advantage that it does not require adversarial fine-tuning. But as shown in **our response to Q1**, ADBM can also handle various unseen attacks, with good generalizability.
>
> Furthermore, compared with LM, ADBM has the following two advantages:
>
> 1) ADBM can perform purification with quite few purification steps, which requires quite fewer inference cost. Here we compared the performance of ADBM and LM under the same inference cost (purification step) against our attack evaluation (PGD200+EOT20, 8/255). We can see that ADBM can have good performance even with one purification step, while LM loses its robustness in this setting.
> |||||
> |-|-|-|-|
> |Method \ Purification steps|1|2|3|
> |LM|13.1|25.0|26.6|
> |ADBM|**45.7**|**47.7**|**47.7**|
>
>
> 2) The evaluation of DiffPure has been extensively studied in recent years, establishing a comprehensive evaluation pipeline and backing it with theoretical guarantees [1, 2], such as certified robustness. In contrast, while LM has been evaluated with white-box attacks, it has not yet been fully supported by theoretical guarantees like certified robustness, which raises concerns about its vulnerability to stronger future attacks. For this reason, in our work, ADBM focuses on further enhancing DiffPure.
>
> Overall, our ADBM constructs an elaborated diffusion bridge from the adversarial distribution to the clean distribution and has a well-defined mathematical structure, where both the forward and reverse processes are Gaussian. This approach fully inherits the desirable mathematical properties of DiffPure. ADBM requires only a few additional lines of code and one epoch of fine-tuning to directly improve DiffPure. We believe ADBM offers an easy-to-use, fine-tuning-based alignment method that improves robustness with affordable computational cost and implementation effort. We will add such discussion to the revised version.
>
>
> **_Q3: About the property of ADBM and the selection of variables such as $\bar{\alpha}$._**
>
> Thank you for your suggestion. We would like to clarify that ADBM retains the property of converging to a standard Gaussian distribution as $t \to T$. ADBM is a new diffusion bridge model, in which **the forward process remains identical to the standard diffusion process and therefore preserves this property**. In the backward process, ADBM builds a Gaussian bridge from the diffused adversarial distribution to the clean distribution. **ADBM modifies only the reverse process of diffusion models** which has been proven to remain Gaussian with a slightly adjusted mean **while the forward process remains unchanged**. Therefore, **ADBM maintains this convergence property of diffusion models**. And thus we can reuse the scheduler ($\bar{\alpha}$) of DDPM without further adjustment.
>
>
>
> **If the reviewer agrees with our clarification above and kindly re-evaluates the value of our work, we would be very grateful. We are happy to address any further questions regarding our work.**
>
>
>
> [1] Carlini N, Tramer F, Dvijotham K D, et al. (Certified!!) Adversarial Robustness for Free! ICLR 2023.
>
> [2] Xiao C, Chen Z, Jin K, et al. Densepure: Understanding diffusion models towards adversarial robustness. ICLR 2023.

---

> > ### Comment · Reviewer_hELf · 2024-11-25
> >
> > I appreciate the clarifications provided and would like to further address the following concerns:
> >
> > 1. While I recognize the evidence that ADBM can generalize to unseen types of attacks, I am interested in its capability to handle **different $\epsilon$ budgets than those used during training**. If it's not feasible to provide experimental results, could you share some insights on how ADBM might withstand unseen attack budgets?
> >
> > 2. **Regarding the parameter settings, LM in RDC uses a step size of 5.** I highly recommend conducting experiments using the same parameters to ensure that the inherent efficacy of LM is not compromised. Raising the purification step of LM from 3 to 5 does not significantly increase the inference cost.
> >
> > 3. As you mentioned, ADBM maintains the same forward process of the diffusion model but alters the reverse process. Given that in diffusion models, **the forward and reverse processes are uniquely corresponding** (as in Score SDE [1] where the reverse SDE is the corresponding inverse of the forward SDE), **is the reverse process of ADBM still corresponds to its forward process correctly**? I am concerned about potential inconsistencies in the forward and reverse process of ADBM and would appreciate further clarification.
> >
> > [1] Score-Based Generative Modeling through Stochastic Differential Equations, ICLR 2021

---

> > > ### Author Response · Authors · 2024-11-25
> > > **Response to the further questions**
> > >
> > > Thank you for the timely and useful feedback. Our further responses are shown below:
> > >
> > > **Q1. About the insights on how ADBM might withstand unseen attack budgets?**
> > >
> > >
> > >
> > > We appreciate your recognition of the evidence that ADBM can generalize to unseen types of attacks. We guess that the attack budgets that ADBM can withstand may depend on two factors:
> > >
> > > 1. The generalization ability of adversarial training. As demonstrated by Liu et al. [1], adversarial training using $\epsilon = 4/255$ under the $L_\infty$ bound can generalize to $\epsilon = 8/255$ under the $L_\infty$ bound, as well as to $L_1$ and $L_2$ cases.
> > > 2. The generalization of diffusion models in handling noise. Diffusion-based purification is not effective against excessive noise that may corrupt most of the information in the images. For instance, when the forward timesteps are set to 200 on CIFAR-10, even the accuracy on clean images cannot exceed 80% [2]. If the attack budget is comparable to the noise introduced by 200 forward timesteps, the robustness of ADBM may be corrupted.
> > >
> > > But overall, such high attack budgets typically make it challenging for even humans to recognize the images.
> > >
> > >
> > >
> > >
> > >
> > > [1] Liu C, et al. A comprehensive study on robustness of image classification models: Benchmarking and rethinking. IJCV, 2024.
> > >
> > > [2] Lee M, Kim D. Robust evaluation of diffusion-based adversarial purification. CVPR, 2023.
> > >
> > >
> > >
> > > **Q2. About the experiments on LM:**
> > >
> > >
> > > Thank you for your follow-up question. In our initial response, we demonstrated that ADBM has significant advantages over LM when the purification steps are set to ≤ 3, under our reliable evaluation using the exact gradient.
> > >
> > > However, as noted in the original paper on LM (**specifically in the Exact Gradient Attack part of Section 4.4**), computing the exact gradient of LM with more purification (optimization) steps incurs a substantial memory cost due to the need for computing second-order derivative, which has also observed by us.  Thus, we apologize that we are unable to provide results for LM with > 3 purification steps under our reliable evaluation with exact gradient, as this leads to out-of-memory errors
> > >
> > > As an alternative, we present the results for both ADBM and LM with 5 purification steps (the original setting for LM) under the BPDA attack on CIFAR-10 ($L_\infty$, 8/255, PGD200+EOT 20, suggested by Reviewer pfC6, see also our responses to Q2.1 of Reviewer pfC6):
> > >
> > > ||||
> > > |-|-|-|
> > > |Method|Clean|Linf|
> > > |LM (5 steps)|85.35|68.55|
> > > |ADBM (5 steps)|**91.93**|**70.51**|
> > >
> > >
> > > We can see that ADBM performed better than LM in this setting. Especially on clean images, ADBM can significantly surpass LM. These results further confirms the advantage of ADBM. But we also note that these results should be cautiously explained as the BPDA is a weak attack for diffusion-based purification compared with the evaluation using exact gradient.
> > >
> > >
> > >
> > > [3] Chen H, et al. Robust Classification via a Single Diffusion Model. ICML, 2024
> > >
> > >
> > >
> > > **Q3. About the  forward and reverse process of ADBM**
> > >
> > > We would like to clarify that **the reverse process of ADBM indeed still corresponds to its forward process correctly**. The distinction between the reverse process of ADBM and that of original diffusion models lies in the different trajectories assumed in the diffusion process. In ADBM, we assume the existence of adversarial noise $\epsilon_a$ at the starting point of the forward process during training (see Section 4.1). Our derivation and design are based on this assumption and strictly follow the paradigm of diffusion models. ADBM only initiates the reverse process from a different starting point compared to original diffusion models and thus there cannot be inconsistency in the forward and reverse process of ADBM. In fact, this is also why we can apply any reverse samplers (including DDPM, DDIM) developed for the original diffusion models without any modification (Section 4.3).
> > >
> > >
> > >
> > > **We hope this further clarification meets your expectations. We would be happy to see the reviewer's further feedback on any additional questions if you still have concerns about our clarifications.**

---

> > > > ### Comment · Reviewer_hELf · 2024-11-27
> > > >
> > > > As we know, the main limitation of adversarial training is its inability to effectively defend against unseen attacks, as it is specifically trained for certain attack types. This limitation is well-documented in works [1, 2]. Additionally, I intended to refer to "different attack budgets from training," not "high attack budgets."
> > > >
> > > > **I disagree that using a diffusion model specifically trained on adversarial examples is a good solution since it may sacrifice the generalizability to unseen attacks.** In contrast to [2, 3], which use diffusion models for purification without training on adversarial examples generated by specific attack methods, ADBM may hurt the generalization ability to unseen attacks, including different attack types, threat models, and attack budgets. Therefore, I find the motivation for this work unconvincing and have decided to maintain my current score.
> > > >
> > > > [1] Perceptual adversarial robustness: Defense against unseen threat models, ICLR 2021
> > > > [2] Robust classification via a single diffusion model, ICML 2024
> > > > [3] Diffusion Models for Adversarial Purification, ICML 2022

---

> > > > > ### Author Response · Authors · 2024-11-28
> > > > > **Response to the further comment of Reviewer hELf**
> > > > >
> > > > > **Dear Reviewer hELf**:
> > > > >
> > > > > Thank you for your further feedback. We regret that you seem to disagree with the use of adversarial training (AT) to enhance adversarial robustness, despite their widespread application in various works aimed at improving the robustness. We would like to respond further as follows:
> > > > >
> > > > >
> > > > >
> > > > > First, while AT has relatively limited generalization to unseen attacks, this does not imply that AT sacrifices generalizability compared to models without AT. In fact, the limited generalization ability is relative to defenses against seen attacks. **When comparing an AT model to a non-AT model, we can clearly see that the AT model increases robustness against various attack types, threat models, and attack budgets.** This is also reflected in the work [1] you referred to, where Table 2 of [1] shows that the model using $\ell_\infty$-based AT significantly outperforms the non-AT model in robustness against several unseen attacks, including $\ell_2$, Stadv, and ReColor, etc, demonstrating **generalizability to unseen attacks**. In [2, 3], we observe further evidence of the scalability of AT, i.e., as models and datasets increase, AT exhibits improved generalization to unseen attacks, as seen in Table 5 of [3]. In addition, the generalization ability of AT to unseen attacks may be further enhanced by advanced AT techniques, including the work you mentioned [1], which suggests that **PAT, still an AT method, can significantly boost performance in generalization to unseen attacks.**
> > > > >
> > > > > Secondly, we have evaluated ADBM on **four unseen attacks** in Tables 4 and A7, including RayS, SPSA, Transfer, and Square attacks. Furthermore, we have conducted experiments on **four additional unseen threats**, including patch-like attacks (PI-FGSM and NCF) and recent diffusion-based attacks (DiffAttack and DiffPGD), with results shown in Table A8. **Under all these unseen attacks, which cover various types of adversarial attacks, ADBM demonstrates better generalization ability than DiffPure [4], which does not utilize AT-like methods. We believe our evaluation on unseen attacks is comprehensive and provides sufficient evidence that ADBM can effectively defend against unseen attacks.** These results about unseen attacks have also **recognized by both reviewer H77b and pfC6**. Regarding [5], we have discussed in detail the advantages that ADBM offers compared to that work in the previous rebuttal.
> > > > >
> > > > > Finally, we would like to emphasize that one of our motivations for ADBM is to *provide "a paradigm for enhancing diffusion-based purification, where we offer detailed proofs to show that ADBM can ultimately cancel the adversarial perturbation term present in adversarial examples"*, as summarized by Reviewer pfC6. **ADBM itself does not assume the form of adversarial noise**, although we need to specify particular noise during training. **The generalization ability of ADBM to unseen attacks may be further enhanced by advanced AT techniques, including the method proposed in [1], which suggests that PAT, still an AT method, can significantly boost performance in generalization to unseen attacks.**
> > > > >
> > > > > We believe that the novel diffusion process of ADBM will be beneficial to the community, from both the AT and AP persepective. **We hope this further clarification address your concens. We would be happy to see the reviewer's further feedback on any additional questions on our clarifications.**
> > > > >
> > > > >
> > > > >
> > > > > [1] Perceptual adversarial robustness: Defense against unseen threat models, ICLR 2021
> > > > >
> > > > > [2] Liu C, et al. A comprehensive study on robustness of image classification models: Benchmarking and rethinking. IJCV, 2024.
> > > > >
> > > > > [3] Wang Z, et al. Revisiting Adversarial Training at Scale.  CVPR. 2024: 24675-24685.
> > > > >
> > > > > [4] Diffusion Models for Adversarial Purification, ICML 2022
> > > > >
> > > > > [5] Robust classification via a single diffusion model, ICML 2024

---

> > > > > > ### Comment · Area_Chair_26y4 · 2024-11-30
> > > > > > **Further discussions?**
> > > > > >
> > > > > > Dear Reviewer hELf,
> > > > > >
> > > > > > Thanks for your time spent on this review. Could you look at the latest response and reply to the authors? Your latest concern might not be very specific/concrete to the authors, and your further comments will be much appreciated.
> > > > > >
> > > > > > Best regards,
> > > > > >
> > > > > > Your Area Chair

---

> > > > > > ### Comment · Reviewer_hELf · 2024-12-02
> > > > > >
> > > > > > Thank you for your response. Regarding your latest feedback, I would like to provide further clarification.
> > > > > >
> > > > > > In my response, I never denied the effectiveness of Adversarial Training (AT). What I pointed out, however, is that prior work [1,2] has shown that the generalization of AT to unseen attacks tends to degrade. This comparison is not between AT and non-AT methods, but rather between AT and purification methods.
> > > > > >
> > > > > > The key advantage of purification over AT is that it does not require adversarial examples generated by a specific attack during training, which allows purification methods to generalize better to unseen attacks. In contrast, ADBM relies on specific adversarial noise during training,  which can potentially hinder the generalization ability of the ordinary diffusion model (such as the one used by DiffPure or LM of RDC). This is why I did not give a higher score.
> > > > > >
> > > > > > [1] Perceptual adversarial robustness: Defense against unseen threat models, ICLR 2021
> > > > > >
> > > > > > [2] Robust classification via a single diffusion model, ICML 2024

---

> > > > > > > ### Author Response · Authors · 2024-12-02
> > > > > > > **Response to the latest comment of Reviewer hELf**
> > > > > > >
> > > > > > > **Dear Reviewer hELf:**
> > > > > > >
> > > > > > > We sincerely thank you for your further feedback and the time spent on the review and discussions. We respect your stance on our paper. We would like to emphasize the following evidence and clarifications again:
> > > > > > >
> > > > > > > 1) From your response, we guess that you have realized the effectiveness of AT in improving robustness against unseen attacks compared to non-AT models. **If this is case, a reasonable deduction is that, compared to a pure AP module without AT, an AP module (diffusion models) combined with AT techniques should exhibit improved robust generalization to unseen attacks instead of hindering the generalization ability.** Our comprehensive experiments have validated this deduction: **ADBM indeed improved the robustness under various unseen attacks compared with other AP methods without AT such as DiffPure**, as detailed in our last response (**the second point**). These improvements regarding unseen attacks have also been recognized by **both Reviewer H77b and Reviewer pfC6**.
> > > > > > >
> > > > > > > 2) Last but not the least, although we used AT-like techniques, we would like to emphasize that one of the key contribution of ADBM is to *provide "a paradigm for enhancing diffusion-based purification, where we offer detailed proofs to show that ADBM can ultimately cancel the adversarial perturbation term present in adversarial examples"*, as summarized by Reviewer pfC6. As also pointed out by Reviewer pfC6, "*an interesting part is that ATOP (*one recent AP combining with AT method [1]*) cannot work for the diffusion models, which further enhances the contribution of this paper"*. **ADBM itself does not assume the form of adversarial noise**, although we need to specify particular noise during training. **The generalization ability of ADBM to unseen attacks may be further enhanced by advanced AT techniques, including the method proposed in [2], which suggests that PAT, still an AT method, can significantly boost performance in generalization to unseen attacks.**
> > > > > > >
> > > > > > > Taken together, We believe that the novel diffusion process of ADBM will be beneficial to the community, from both the AT and AP persepective. **We hope this further clarification address your concens. We would be happy to see the reviewer's further feedback on any additional questions on our clarifications.**
> > > > > > >
> > > > > > > [1] Adversarial Training on Purification (AToP): Advancing Both Robustness and Generalization. ICLR, 2024
> > > > > > >
> > > > > > > [2] Perceptual adversarial robustness: Defense against unseen threat models, ICLR 2021

---

### Comment · Area_Chair_26y4 · 2024-11-22

Dear Authors and Reviewers,

The discussion phase has passed 10 days. If you want to discuss this with each other, please post your thoughts by adding official comments.

Thanks for your efforts and contributions to ICLR 2025.

Best regards,

Your Area Chair

---

### Author Response · Authors · 2024-11-24
**Hoping for further feedback**

**Dear reviewers**,

We thank you again for the valuable and constructive comments. We are looking forward to hearing from you about any further feedback.

If you find our response satisfactory, we hope you might view this as a sufficient reason to reconsider the rating further.

If you still have questions about our paper, we are willing to answer them and improve our manuscript.

Best, Authors

---

### Meta-Review · Area_Chair_26y4 · 2024-12-21

**Metareview:**

This paper makes several contributions to the field: 1) Propose the full gradient evaluation for diffusion models based on AP. 2) Propose an AT-based paradigm to enhance diffusion models based-AP, where the author offers detailed proofs to show that AT could finally cancel the adversarial perturbation term hidden on the adversarial samples. 3) The author reports detailed experiments, including unseen attacks, to prove the validity and efficiency of the proposed method. Although the proposed method might introduce additional training cost to diffusion model-based AP, its overall contribution is enough.

**Additional Comments On Reviewer Discussion:**

All technical concerns are addressed during the rebuttal. One reviewer questions the AT-based framework cannot be used in transferred attacks. However, there is no theoretical guarantee to make that claim and we can only empirically show the performance under the transfer attacks. The authors indeed did related experiments to show the effectiveness of the proposed method. Thus, the concern is resolved.

---

### Decision · Program_Chairs · 2025-01-22

Accept (Poster)